# iQSP: A Sequence-Based Tool for the Prediction and Analysis of Quorum Sensing Peptides via Chou’s 5-Steps Rule and Informative Physicochemical Properties

**DOI:** 10.3390/ijms21010075

**Published:** 2019-12-20

**Authors:** Phasit Charoenkwan, Nalini Schaduangrat, Chanin Nantasenamat, Theeraphon Piacham, Watshara Shoombuatong

**Affiliations:** 1College of Arts, Media and Technology, Chiang Mai University, Chiang Mai 50200, Thailand; phasit.c@cmu.ac.th; 2Center of Data Mining and Biomedical Informatics, Faculty of Medical Technology, Mahidol University, Bangkok 10700, Thailand; nalini.sch@mahidol.edu (N.S.); chanin.nan@mahidol.edu (C.N.); 3Department of Clinical Microbiology and Applied Technology, Faculty of Medical Technology, Mahidol University, Bangkok 10700, Thailand; theeraphon.pia@mahidol.ac.th

**Keywords:** quorum sensing peptides, physicochemical properties, support vector machine, random forest, machine learning, classification

## Abstract

Understanding of quorum-sensing peptides (QSPs) in their functional mechanism plays an essential role in finding new opportunities to combat bacterial infections by designing drugs. With the avalanche of the newly available peptide sequences in the post-genomic age, it is highly desirable to develop a computational model for efficient, rapid and high-throughput QSP identification purely based on the peptide sequence information alone. Although, few methods have been developed for predicting QSPs, their prediction accuracy and interpretability still requires further improvements. Thus, in this work, we proposed an accurate sequence-based predictor (called iQSP) and a set of interpretable rules (called IR-QSP) for predicting and analyzing QSPs. In iQSP, we utilized a powerful support vector machine (SVM) cooperating with 18 informative features from physicochemical properties (PCPs). Rigorous independent validation test showed that iQSP achieved maximum accuracy and MCC of 93.00% and 0.86, respectively. Furthermore, a set of interpretable rules IR-QSP was extracted by using random forest model and the 18 informative PCPs. Finally, for the convenience of experimental scientists, the iQSP web server was established and made freely available online. It is anticipated that iQSP will become a useful tool or at least as a complementary existing method for predicting and analyzing QSPs.

## 1. Introduction

It is widely known that bacteria can interconnect within its population using cell–cell communication tools. One such tool, quorum sensing (QS) is a molecular mechanism that depends on the population density to trigger cell–cell signaling which changes the behavior of the bacterial community when the population reaches a critical level [1,2]. Genes directing the beneficial activities performed synchronously by a bacterial population are controlled by the QS mechanism (i.e., bioluminescence, sporulation, competence, antibiotic production, biofilm formation, and virulence factor secretion) [3]. Furthermore, in order to orchestrate these collective behaviors, QSPs (quorum sensing peptides or autoinducing (AI) peptides) are secreted by the bacteria to respond to extracellular signaling molecules. Typically, in a fresh culture, bacteria continually generate the signal starting at a low concentration, which accumulates as the population density increases. Upon reaching a threshold concentration, receptor protein interaction occurs which activates receptor kinase by phosphorylation thus, inducing a coordinated change in gene expression via transcription of target genes in the population [4]. QSPs have been widely studied in Gram-positive bacteria and are shown to be species specific (i.e., *Staphylococcus* spp., *Clostridium* spp., or *Enterococcus* spp) [5]. On the other hand, Gram-negative bacteria (i.e., *Pseudomonas* spp., *Acinetobacter* spp., or *Burkholderia* spp.) reportedly produce a different class of autoinducers known as, acyl-homoserine lactones (AHLs) which comprise of a lactone ring coupled with an aliphatic acyl chain with varying length and modifications [6]. In addition, a wide-ranging variety of other signaling molecules have also been identified [7], including fatty acids used by *Xanthomonas* spp., *Burkholderia* spp., *Xylella* spp. [8], ketones (*Vibrio* spp. and *Legionella* spp. [9]), epinephrine, norepinephrine and AI-3 (enterohemorrhagic bacteria; [10]) or quinolones (*Pseudomonas aeruginosa*; [11]). Additionally, AI-2, a furanosyl borate diester, is used by both Gram-negative and Gram-positive bacteria [12]. Finally, QSPs are well understood at the molecular level in many bacterial species, and have been extensively reviewed [13,14,15,16,17]. An example of experimentally elucidated QSP structures are shown in Figure 1.

As previously mentioned, QSPs control the production of virulence factors in bacteria including many antibiotic resistant bacteria such as lectin, exotoxin A, pyocyanin, and elastase in *Pseudomonas aeruginosa* [15,18,19] and hemolysins, protein A, enterotoxins, lipases, and fibronectin proteins in *Staphylococcus aureus* [20,21]. These virulence factors allow the bacteria to evade the host immune responses. In addition, the overuse and abuse of antibiotics coupled with increased resistance, has prompted the discovery of various anti-QS agents as alternatives to traditional antibiotics [4]. These anti-QS agents can abolish the QS signaling pathways and prevent the formation or accumulation of virulence factors, therefore reducing bacterial virulence without causing drug-resistance. However, no such drug in currently approved by the FDA. At present, the combination of anti-QS agents with antibiotics provides the most effective strategy to combat bacterial infections [22,23]. As such, many studies have successfully demonstrated the synergistic effects of combining antibiotics with anti-QS agents [24,25,26,27]. Thus, it is promising to discover novel therapeutic agents to help boost research in this area. 

Although, experimental approaches are known as an objective way to identify the biological activities of QSPs, it is time-consuming and costly. Meanwhile, with the avalanche of the peptide sequences in many free-access databases, the development of fast, efficient, and intelligent computational models for predicting and analyzing QSPs is urgently need to serve clinical application, drug development, and basic research. Currently, there are only two computational methods [28,29] developed for discriminating QSPs from Non-QSPs using the information of peptide sequences as summarized in Table 1. The first QSP predictor was proposed by Akanksha et al. [29] named QSPpred. This method was developed by using support vector machine (SVM) cooperating with four types of peptide features, i.e., amino acid composition (AAC), dipeptide composition (DPC), binary pattern of N- and C-terminal residues and physicochemical properties (PCP). In 2018, Leyi et al. proposed a second QSP predictor named QSPpred-FL. This predictor was a two-layer prediction framework, where the first layer was used to establish initial features derived from 10 different types of peptide features and generate a new feature vector having 99-dimensional feature vector, while the final QSP predictor (QSPpred-FL) was construct in the second layer by using random forest (RF) with the top four features consisting of g-gap dipeptide composition (GDP), overlapping property features (OVP), composition–transition–distribution (CTD) and adaptive skip dipeptide composition (ASDC). Although, the two above-mentioned methods could provide quite promising prediction results, there is still room for improvement in the performance and interpretability of the predictors. First, QSPpred and QSPpred-FL were constructed with 630D and 913D feature vectors, respectively. The prediction model constructed with high-dimensional feature spaces could be prone to overfitting and might yield overestimated prediction results [30,31,32,33]. Second, QSPpred-FL was not evaluated on an independent set therefore, its predictive ability on unknown peptides could not be determined. Third, QSPpred and QSPpred-FL was performed on the benchmark and independent datasets with a single random sampling procedure. Hence, these two methods might tend to find the possible bias of the random sampling process and provided a good predictive result by chance. Finally, the mechanism of QSPpred and QSPpred-FL suffers from their low interpretable ability for experimental and related researchers as it is not easy to identify and assess which features provide crucial contribution to the high prediction results.

To improve the prediction and interpretability performances, we proposed a systematic effort via Chou’s 5-steps rule [34,35] for predicting and analyzing QSPs called the iQSP. The development of iQSP consists of five main parts: (i) Collecting the benchmarking dataset, (ii) representing a peptide sequence with PCPs from the AAindex database, (iii) selecting m informative PCPs from 531 PCPs, (iv) developing the QSP predictor using SVM model with the m informative PCPs, and (v) extracting interpretable rules by using the RF method with the m informative PCPs called IR-QSP. In this study, the m informative PCPs were identified using the SVM model in cooperating with the genetic algorithm utilizing self-assessment-report (GA-SAR). The performance comparisons showed that iQSP achieved an accuracy, MCC and auROC of 93.00%, 0.91% and 0.96%, respectively, as assessed by the rigorous independent validation test, which showed a significant improvement as compared with QSPpred. Amongst a set of interpretable rules, there were seven out of eight interpretable rules that could yield a prediction accuracy of greater than 80%. Finally, iQSP was developed as a user-friendly and publicly accessible web server that allow robust predictions to be made without the need to develop in-house prediction models.

## 2. Results and Discussion

In this study, we developed a sequence-based model for predicting and analyzing QSPs named iQSP. Firstly, the sequence logo representation and Gini index of amino acids were used for characterizing the informative properties between QSPs and Non-QSPs. Secondly, the GA-SAR algorithm was used to generate ten feature subsets, where each subset consisted of m informative features of PCPs. Thirdly, SVM models were individually constructed using the mentioned feature subsets. After that, the if–then interpretable rules (called IR-QSP) were generated by using the RF method and the optimal feature subset. Finally, iQSP is deployed as a free prediction web server so as to afford easy and rapid classification of query protein sequence as being QSPs and Non-QSPs. The overall framework of the proposed model, iQSP is shown in Figure 2.

### 2.1. Composition Analysis

The analysis of feature importance can provide valuable information for predicting its function and activity. Previously, AAC has been used for analyzing the inherent characteristics and patterns of many therapeutic peptides [36,37,38,39,40,41] and protein functions [42,43,44]. In this study, the mean decrease of Gini index (MDGI) was utilized to rank the importance of each AAC feature. Features with the largest MDGI value are considered to be important as they significantly contribute to the prediction performance. In order to increase the reliability and validity for determining the feature importance, 100 RF models were constructed by varying the mtry parameter settings from 1 to 100 (mtry = 1, 2, 3, …, 100) and fixing the ntree parameter with 500 [36,39,42,45]. Finally, the average value of MDGI on 100 runs of feature importance estimations were used in this study. Table 2 lists the percentage values of the twenty amino acids for both QSPs and Non-QSPs along with amino acid compositional differences between the two classes as well as their MDGI values. Furthermore, the sequence logo [46] of the first and last ten residues at the N- and C-terminal regions of both QSPs and Non-QSPs were created to visualize the positional information for each amino acid as shown in Figure 3.

Table 2 shows that the ten top-ranked important amino acids according to MDGI values are Phe, Lys, Leu, Val, Trp, Ile, Ala, Tyr, Arg, and Cys. There are 6 out of 10 top-ranked important amino acids having an MDGI value larger than 10, i.e., Phe, Lys, Leu, Val, Trp, and Ile. As seen, amongst the ten informative amino acids, the analysis of AAC with the percentage of certain residues for QSPs suggests that Phe, Trp, and Tyr are dominant in QSPs, while Lys, Leu, Ala, Arg, and Cys are dominant in Non-QSP peptides at a significance level of *p*-value ≤ 0.05. Furthermore, the four sequence logo representations were created to visualize the positional information for each amino acid as shown in Figure 3. The overall stack height of each position indicates its sequence conservation while the size of the residue represents its propensity. Figure 3a,b reveal that Ser, Gly, Phe, and Arg as well as Phe and Lys are abundant in the first ten residues from the N- and C-terminal regions, respectively, of QSPs. Meanwhile, Figure 3c,d show that Met, Leu, Lys, Ala, and Gly as well as Leu and Cys are abundant in the first ten residues from the N- and C-terminal regions, respectively, of Non-QSPs. Therfore, the information gathered from the analysis of feature importance and the sequence logo illustrations in Table 2 and Figure 3, respectively, can be briefly summarized as follows: (i) Phe, Lys, Leu, Ala, and Arg are crucial amino acid residues that could potentially be used for discriminating QSPs from Non-QSPs; (ii) Phe and Ser as well as Lys and Leu are seen to be favored by QSPs and Non-QSPs, respectively, which are found in the first and last ten residues at the N- and C-terminal regions; and (iii) These observations were in good consistency with the feature importance as estimated using MDGI values where Phe, Lys, Leu and Ser are ranked 1, 2, 3, and 11, respectively. Moreover, our analysis were quite compatible with previous works [29,47].

### 2.2. Prediction Capabilities of the Different Subset of Physicochemical Properties

To make a fair comparison with the existing methods, a series of comparative experiments were carried out, while the same benchmark (200 QSPs and 200 Non-QSPs) and independent (20 QSPs and 20 Non-QSPs) datasets were used to develop and investigate the efficiency and effectiveness of our proposed QSP predictor. However, previous studies [28,29,48] performed these two datasets using a single random sampling procedure. As elaborated in [39,40,42,43,45,49,50], this procedure might find a possible bias of the random sampling process and provided a good predictive result by chance. Therefore, we repeated this construction procedure with ten independent rounds to alleviate the aforementioned problems [36,39,41,42,43,45]. In this study, the proposed GA-SAR algorithm was used to identify m informative features from 531 PCPs, where the number of m is in the range of 5–20. We hypothesize that if a feature is selected by GA-SAR, it is considered to be beneficial for QSP prediction [51,52,53]. Due to the non-deterministic characteristics of the GA-SAR algorithm, ten individual experiments were performed to generate ten different feature subsets. The lists of m informative features in the ten different feature subset are demonstrated in Table A1. In our experimental setting, each feature subset was used as the input feature to construct 10 SVM models based on random sampling with ten independent rounds. Therefore, the final prediction results of 10-fold CV and independent validation tests of each feature set were obtained by averaging the five statistical parameters (Ac, Sn, Sp, MCC, and auROC) on the benchmark and independent datasets, as shown in Table 3 and Table 4, respectively.

Amongst the ten different feature subsets, Table 3 shows that the subsets with the five highest Ac over 10-fold CV are subsets 10, 5, 4, 3, and 1 (92.19 ± 1.09%, 92.01 ± 1.40%, 91.63 ± 1.75%, 91.58 ± 1.77%, and 91.23 ± 1.31%, respectively). Meanwhile, the subsets with the five highest Ac over the independent validation test are subset 6, 10, 5, 1, and 4 (93.00 ± 1.97%, 92.50 ± 1.67%, 92.50 ± 2.36%, 92.50 ± 1.67%, and 92.25 ± 2.19%, respectively), as summarized in Table 4. As noticed in Table 3 and Table 4, subsets 10, 5, 4, 1, and 6 showed good predictive powers with their ranks (10-fold CV, independent validation test) at (1, 2), (2,3), (3, 5), (5, 4), and (6,1) respectively. Although, subset 6 having a 91.07 ± 1.77% Ac was ranked at 5, its prediction result over independent validation test with 93.00 ± 1.97% Ac outperformed that of other subsets. Due to the fact that the independent test is an effective way to demonstrate the robustness and reliability of the model in real-world applications [36,39,40,41,42,43,45,49,50], it could be stated that subset 6 having eighteen informative PCPs provided a crucial contribution to the prediction performance. This feature subset yielded a prediction performance (Ac/MCC/auROC) over the 10-fold CV and independent validation test of 91.07 ± 1.77%/0.82 ± 0.04/0.91 ± 0.10 and 93.00 ± 1.97%/0.86 ± 0.04/0.96 ± 0.02, respectively.

Furthermore, the performance comparisons between SVM models with 531 PCPs as well as eighteen informative PCPs were conducted to investigate the effectiveness of our selected feature subset, which is illustrated in Figure 4. As noticed in Figure 4c, the values of Ac, Sn, Sp, MCC, and auROC derived from using the eighteen informative PCPs are higher than using the 531 PCPs by 7%, 2%, 11%, 8%, and 3%, respectively. These results demonstrated that the inclusion of numerous redundant and uninformative features caused poor prediction results. For convenience, the best QSP predictor based on SVM model in conjunction with the eighteen informative PCPs will be referred to as iQSP.

### 2.3. Comparison with Existing Methods

To demonstrate the effectiveness and power of our method, we conducted a comparative study of our final model (named iQSP) with the existing methods. To date, there are only two existings methods developed for the prediction of QSPs, i.e., QSPpred [29] and QSPpred-FL [28,48], performing on the benchmark and independent datasets over 10-fold CV and independent validation test. Table 5 lists the preformance comparisons of iQSP and the existing methods. From Table 5, we can observe that QSPpred-FL yields the highest prediction performance of 94.30% Ac and 0.885 MCC over 10-fold CV, while our proposed model iQSP gave a 91.07 ± 1.77% Ac and 0.82 ± 0.04 MCC. On the other hand, based on the independent validation test, iQSP outperformed that other methods with 93.00 ± 1.97% Ac, 0.86 ± 0.04 MCC and 0.96 ± 0.02 auROC, which was better than the existing QSP predictors [28,29,48]. Although, iQSP achieved slightly better than QSPpred-FL, our proposed model showed significant improvement than QSPpred-FL considering the two objectives: using the less complexity of prediction methods (1 SVM vs. 99 RFs) and a minimum number of features used (18D vs. 913D).

To further investigate the power of the proposed iQSP, we compared its performance with other conventional classifiers, i.e., k-nearest neighbor (k-NN), decision tree (rpart), and random forest (RF). The k-NN, rpart and RF classifiers were performed on the same datasets and implemented using the caret R package [54]. Rigorous 10-fold CV and independent validation test with ten independent rounds of these classifiers based on the optimal feature subset are reported in Table 6 and Figure 5. The more details of the parameter optimization of these three classifiers were described in the works [37,38,55,56,57,58,59,60,61]. Based the independent validation test, we noticed that the Ac, MCC and auROC values of iQSP were higher than those of other classifiers by >2%, >4%, and >2%, respectively, suggesting that iQSP holds very high potential to provide an accurate and reliable result in unseen peptides when compared to the existing methods and the conventional classifiers developed in this study.

Based on Table 3, Table 4, Table 5 and Table 6 and Figure 3, the superior performance of our proposed model iQSP over 10-fold CV and independent validation test might mainly be due to the following reasons: (i) Performing with multiple random sampling procedure to protect against the risk of having good predictive result by chance [39,40,41,42,43,49,50]; (ii) using an efficient feature selection method (GA-SAR) to identify m informative features from 531 PCPs. Using eighteen informative PCPs could provide faster and more cost-effective models, while model developers could gain an insight into the underlying prediction processes [58,62,63,64]; (iii) selecting a powerful method for QSP prediction. Although, iQSP displayed a superior performance over the existing methods assessed by the rigorous cross-validation methods, there is still room for further improvements, including increasing the size of QSPs by gathering peptide sequences from various data sources, utilizing an interpretable learning algorithm, such as scoring card method [44,53], improving the interpretation of important features responsible for the biological activity [50,64] and exploring different ML algorithms, such as extreme gradient boosting [65] or deep learning [66].

### 2.4. Feature Contribution Analysis

Informative features play a vital role for developing an accurate predictor as well as providing a better understanding of QSPs. In this section, the eight informative PCPs utilized to analyze and characterize QSPs are explored. In addition, the MDGI score was used to rank the importance of those informative PCPs. Table 6 summarizes the detailed information of the eighteen informative PCPs and their corresponding MDGI scores. As noticed in Table 7, the top seven important PCPs are QIAN880137, AURR980102, ROBB760113, PRAM820101, GRAR740101, PALJ810111, and PONP800102 having an MDGI value of greater than 10. The most important PCP is the AAindex ID QIAN880137 with an MDGI value of 32.50 denoting ‘weights for coil’. In the analysis, two out of the eighteen informative PCPs are related to AAC and composition, i.e., DAYM780101 (MDGI = 8.64) and GRAR740101 (MDGI = 12.26). Akanksha et al. [29] reported that using AAC and DPC as input features yield Ac values as high as 89.00% and 87.50%, respectively, as evaluated by the independent validation test. Furthermore, amongst the eighteen informative PCPs, there are two informative PCPs, i.e., PONP800102 (MDGI = 10.96) and MANP780101 (MDGI = 8.36), related to hydrophobicity. In 2007, Raymond et al. [47] tested their hypothesis that the hydrophobic face of 21-amino-acid signaling peptide might be important for receptor binding by replacing a hydrophobic residue (Phe) with a hydrophilic residue (Gln). In this study, three peptides, i.e., F7Q, F11Q, and F15Q, were synthesized and assessed for their abilities to activate quorum sensing. Their results indicated that the substitution of Phe with Gln significantly affected the activity of the signal peptide in activation of quorum sensing. Additionally, Akanksha et al. [29] also mentioned the importance of Phe in QSPs whereby Phe was dominant in the first and last 5 residues at N- and C-terminal regions. As noticed in Table 2, these results are consistent with our analysis results which show that Phe and Gln are ranked at 1 and 20, respectively. 

### 2.5. Interpretable Rules Acquisition

In this work, the if–then interpretable rules called IR-QSP were constructed by using the RF method in conjunction with the optimal subset consisting of 18 informative PCPs (Table 7), as mentioned above. The main advantages of these constructed rules are twofold: (i) To demonstrate which PCP or which combination of PCPs are effective for QSP prediction, and (ii) to simply discriminate QSPs from Non-QSPs without the need to go through the mathematical and computational details. Table 8 and Table A2 list eight interpretable rules that were important for QSP and twelve that were important for Non-QSP. If a query peptide meets all of the criteria in at least one of the eight rules, then it is identified as QSP. As observed in Table 8 and Table A2, almost all the rules can yield a prediction accuracy of greater than 80%, except for rule #7. Interestingly, these four rules can achieve an Ac value of greater than 90%, i.e., rules #1, #2, #3, and #4. Thus, these results indicate that the construction of rules are reliable and easy-to-use, both in terms of their accuracy and interpretability for predicting and characterizing QSPs.

### 2.6. iQSP Web Server

In an effort to maximize the full potential usage of the predictive model proposed in this study, the model along with optimal parameter settings were embedded inside an R powered website by means of the Shiny package. The resulting iQSP web server is publicly available at http://codes.bio/iqsp/. The server accepts an input the query peptide sequence in FASTA format that it submits for feature calculation and further fed into the predictive model for prediction of the class label as to whether it is a QSP or Non-QSP. Screenshots of the iQSP web server are shown in Figure 6.

The procedure for using the iQSP web server for predicting the class label can be summarized as follows:

Step 1. Proceed to the iQSP web server by going to the URL, http://codes.bio/iqsp/ and wait until the text box below the “Status/Output” (found on the right hand side) returns the message [1] “Server is ready for prediction”.

Step 2. Enter the query peptide sequence (in FASTA format) into the text box found below the text “Enter your input sequence(s) in FASTA format”. Alternatively, the user can also save the FASTA sequences into a text file and upload this text file to the server by clicking on the “Choose File” found below the text “or upload file”.

Step 3. Finally, click on the “Submit” button to start the prediction process. Shortly after, the prediction results will be displayed in the text box found below the “Status/Output”. This will return the prediction results as a 4 column verbatim text for which the first, second, third and fourth column corresponds to an arbitrary sequence identification number (1, 2, 3, etc.), predicted class label (i.e., as either QSP or Non-QSP), the probability score of the query peptide being a Non-QSP and the probability score of the query peptide being a QSP, respectively.

## 3. Materials and Methods

In order to establish a robust sequence-based tool for modeling the investigated QSPs, we followed Chou’s five-step guidelines as mentioned in a series of recent publications [67,68,69,70,71,72] and summarized in two comprehensive review papers [34,35]: (i) Compilation of a reliable dataset that contains experimentally validated sequences for training and validating the model; (ii) quantifying peptides sequences to describe their physicochemical properties; (iii) developing the prediction model using robust algorithm; (iv) assess the prediction model using standard cross-validation tests; and (v) constructing a user-friendly web-server for obtaining the prediction without the need to understand complex mathematical and statistical details. Furthermore, Figure 2 shows the workflow of iQSP which works in discriminating peptides as QSPs or Non-QSPs.

### 3.1. Benchmark Dataset

To make a fair comparison with the existing methods, the same benchmark and independent datasets derived from the work [29] were taken to develop and validate the proposed model. In this study, the benchmark dataset consists of 200 QSPs and 200 Non-QSPs, while the independent dataset consists of 20 QSPs and 20 Non-QSPs. The benchmark (*S_TR_*) and independent (*S_TS_*) datasets used in this study can be summarized by the following formula:
(1)$$S_{TR}=S_{TR}^{+}\cup S_{TR}^{-}$$
(2)$$S_{TS}=S_{TS}^{+}\cup S_{TS}^{-}$$
where $S_{}^{+}$ and $S_{}^{-}$ represent peptide sequences of QSPs and Non-QSPs, respectively, while the symbol ∪ represents the union from the set theory. However, these two datasets were constructed with a single random sampling procedure. Therefore, to alleviate the impact of the random sampling procedure, we repeated this construction procedure with ten independent rounds and the final prediction results were obtained by averaging the five statistical parameters.

### 3.2. Feature Representation

PCP is one of the most intuitive features associated with biophysical and biochemical reactions and is also referred to as an available, easy and interpretable feature. In fact, a total of 531 PCPs without NA values were derived from version 9.0 of the Amino acid index database (AAindex) [58], which is a collection of the published literature pertaining to different physicochemical and biophysical properties of amino acids and pairs of amino acids (http://www.genome.jp/aaindex/). Each PCP consisted of a set of 20 numerical values for amino acids. The PCP feature has been extensively used for the prediction and analysis of various protein [42,52,53] and peptide [36,39,40,41,45] functions. To utilize PCP features for extracting a peptide sequence, peptide with the length of L amino acid residues is encoded into an L-dimensional vector of 531 PCPs (531D). In this study, the number of L is in the range of 5–20, i.e., 5, 6, 7, …, 20. Additional details on how to obtain a minimum number of L can be found in the sub-section of Identification of Informative Physicochemical Properties.

### 3.3. Support Vector Machine

SVM method is an effective ML algorithm for supervised pattern and has been widely used in various biological problems [36,37,38,39,41,43,45,52,73,74,75,76,77,78,79,80]. This method is based on the Vapnik–Chervonenkis theory of statistical learning [81,82,83]. The basic idea of this method is to map the original feature vectors having p-dimensional vector into a higher Hilbert space with n-dimensional vector, where p < n, and then determine a separate hyper plane with the largest distance between the two classes. In this work, each sample on the benchmark ($S_{TR}$) and independent ($S_{TS}$) datasets have a corresponding label (−1 and 1) where +1 and −1 represent QSP and Non-QSP, respectively. In this study, the *kernlab* R package [84] was used to implement the SVM model. To enhance the performance of the SVM model, the regularization parameter *C* and kernel parameter γ were tuned by using grid search method with a cross-validation technique, of which the search space for *C* and γ are [2^−8^, 2^8^] and [2^−8^, 2^8^] with steps of 2 and 2, respectively.

### 3.4. Performance Evaluation

In statistical predictions, there are three testing methods most often used to assess the predictive ability of the model: (i) Sub-sampling test (2-, 5- or 10-fold cross-validation), (ii) jackknife test also known as leave-one-out cross-validation (LOO-CV), and (iii) independent (or external) validation test. The sub-sampling test is one of the most popular cross-validation methods to assess the predictive capability of a model. As described in [35,85] and investigated by Equation (50) of [86], among those testing methods, the jackknife test is considered as one of the most rigorous that can provide a unique result for a given benchmark dataset. However, to perform a fair comparison with the existing methods and reduce the computational time, the 10-fold cross-validation (10-fold CV) and independent validation test were used to evaluate the prediction performance of our models. The former set (10-fold CV) makes use of data from the training set where the data set is separated into ten subsets. Practically, one subset from a total of ten subsets is left out as the testing set while the remaining are used for training the model. This process is repeated iteratively until all data samples have had the chance to be left out as the testing set. 

In order to evaluate the prediction ability of the model, the following sets of four metrics are used:(3)$$\mathrm{Ac}=\frac{\mathrm{TP}+\mathrm{TN}}{\left(\mathrm{TP}+\mathrm{TN}+\mathrm{FP}+\mathrm{FN}\right)}$$
(4)$$\mathrm{Sn}=\frac{\mathrm{TP}}{\left(\mathrm{TP}+\mathrm{FN}\right)}$$
(5)$$\mathrm{Sp}=\frac{\mathrm{TN}}{\left(\mathrm{TN}+\mathrm{FP}\right)}$$
(6)$$\mathrm{MCC}=\frac{\mathrm{TP}\times \mathrm{TN}-\mathrm{FP}\times \mathrm{FN}}{\sqrt{\left(\mathrm{TP}+\mathrm{FP}\right)\left(\mathrm{TP}+\mathrm{FN}\right)\left(\mathrm{TN}+\mathrm{FP}\right)\left(\mathrm{TN}+\mathrm{FN}\right)}}$$
where Ac, Sn, Sp, and MCC represent accuracy, sensitivity, specificity and Matthews coefficient correlation, respectively. TP, TN, FP, and FN represent the instances of true positive, true negative, false positive, and false negative, respectively. Moreover, in order to evaluate the prediction performance of models using threshold-independent parameters, receiver operating characteristic (ROC) curves were plotted using the pROC package in the R software [87]. The area under the ROC curve (AUC) was used to measure the prediction performance, where AUC values of 0.5 and 1 are indicative of perfect and random models, respectively.

### 3.5. Identification of Informative Physicochemical Properties

In order to determine a minimal number of informative features while maximizing prediction accuracy of SVM model, the customized implementation of genetic algorithm (GA), called genetic algorithm utilizing self-assessment-report (GA-SAR), was developed to elucidate the mentioned problem by taking an advantage of an inheritable GA [88] to select an informative feature. GA-SAR utilizes a self-assessment-report (SAR) approach to construct a profile used for reporting the usefulness of 531 PCPs based on the assumption that a good feature will be highly correlated with the output variable, but uncorrelated to each other [51,89]. Before starting the GA-SAR process, the usefulness of 531 PCPs are calculated. After that, the profile is applied in a mutation function of GA to create a chance of adding/deleting features. During the process of GA optimization, features having low usefulness scores are deleted, while a minimal number of informative PCPs are specified. 

The chromosome of GA-SAR consisted of binary genes indicating an occurrence of each feature in the feature set and the parameter of model genes for tuning the parameters of the classifier. Herein, the binary genes contain 531 genes and two 4-bit for encoding the parameters C (2^−8^, 2^−7^, …, 2^8^) and γ (2^−8^, 2^−7^, …, 2^8^) of the SVM model. The identification of informative PCPs using GA-SAR algorithm is described as follows:

(1) Randomly generate 50 chromosomes with randomly assigned values of binary genes to make the number of features (m) equal to our preferred number, where m is in the range from 5 to 20. (Initialization)

(2) Assess the prediction performances for each chromosome over 10-fold CV procedure. (Evaluation)

(3) Implement a tournament selection to prepare a mating pool. (Selecting)

(4) Perform a 20-point crossover on the selected parents. (Crossover)

(5) Apply the SAR mutation operator and if the number of chosen genes are greater than the specified number of features, delete some genes. In contrast, if the number of chosen genes are less than the specified number of features, add some genes. The probability of genes to be added or deleted is referred to as SAR. (Mutation)

(6) Stop if the number of generation is equal to 50; otherwise go to Step 2. (Termination)

### 3.6. Construction of Interpretable Rules

This work presents an interpretable rule extraction of QSPs (IR-QSP) based on the RF method cooperating with the *m* informative PCPs for determining the biophysical and biochemical properties of QSPs. A set of rules from an individual tree is derived from the root to the leaves. In this study, only 100 decision trees were used to extract the if–then interpretable rules for explaining the prediction results by means of RF method, inTrees and xtable packages in the R software [90,91,92]. More details of the rule extraction process can be found in previous related works [36,61,93].

### 3.7. Construction of the iQSP Web Server

The best performing model described in this study is used as the basis for deployment as a web server. Particularly, the underlying machine learning model is encapsulated inside a website by means of the Shiny package in an R programming environment. By default of the Shiny package, the user interface follows a responsive web design principle in which the website can display optimally in various device platform whether it be a mobile phone, tablet or desktop computers of various screen resolutions. The code and data used to operate the iQSP web server is hosted on Digital Ocean and is publicly available at http://codes.bio/iqsp/. In fact, user-friendly and publicly accessible web-servers that can display the findings manipulated by users according to their need might significantly enhance their impacts, driving medicinal chemistry into an unprecedented revolution [34,94,95,96,97,98,99,100,101,102,103,104,105,106,107]. Keeping this point in our mind, we shall make efforts in our future work to provide a web-server to provide such functionality.

## 4. Conclusions

An accurate tool for predicting and analyzing QSPs is essential for understanding their roles in clinical applications and providing a promising way to combat bacterial infections. Thus, in this study, we developed an efficient and interpretable sequence-based predictor for predicting and analyzing QSPs, called iQSP, by utilizing a set of m informative physicochemical properties (PCPs) in conjunction with a powerful support vector machine (SVM). Performance comparisons for both rigorous cross-validation and independent validation tests demonstrated the superiority of iQSP over the existing QSP predictors. Moreover, feature selection and interpretable rule extraction were carried out to construct easy-to-use if–then rules (IR-QSP) for discriminating QSPs from Non-QSPs by using random forest (RF) model and the m informative PCPs. Finally, to help potential users of iQSP, a web server named iQSP was implemented and made freely available online at https://codes.bio/iqsp/. It is anticipated that iQSP may become a powerful and cost-effective approach for predicting and analyzing peptides on a large scale. Due to various potential application of our systematic approach employed in this study, we could extend this for predicting and analyzing many other types of protein and peptide functions such as *S*-palmitoylation sites in proteins [68], lysine crotonylation sites [108], and phosphotyrosine sites [109]. Furthermore, our method could be integrated with other beneficial peptide features such as pseudo amino acid composition [110,111,112] or amphiphilic pseudo amino acid composition as proposed [113] by Chou [35,114] for further improving the QSP prediction

## Figures and Tables

**Figure 1 ijms-21-00075-f001:**
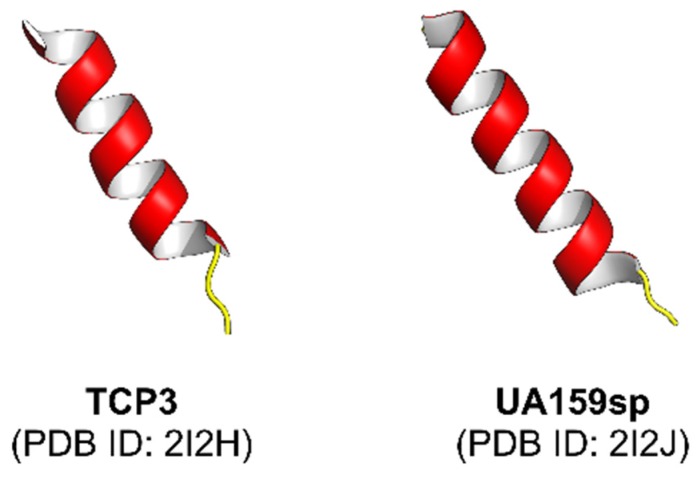
Structures of selected quorum sensing peptides that have been experimentally elucidated, where red and yellow colors represent alpha-helix and loop structures, respectively. Each structure is labelled by a common name followed by the Protein Data Bank identification number (PDB ID) in parenthesis on the subsequent line.

**Figure 2 ijms-21-00075-f002:**
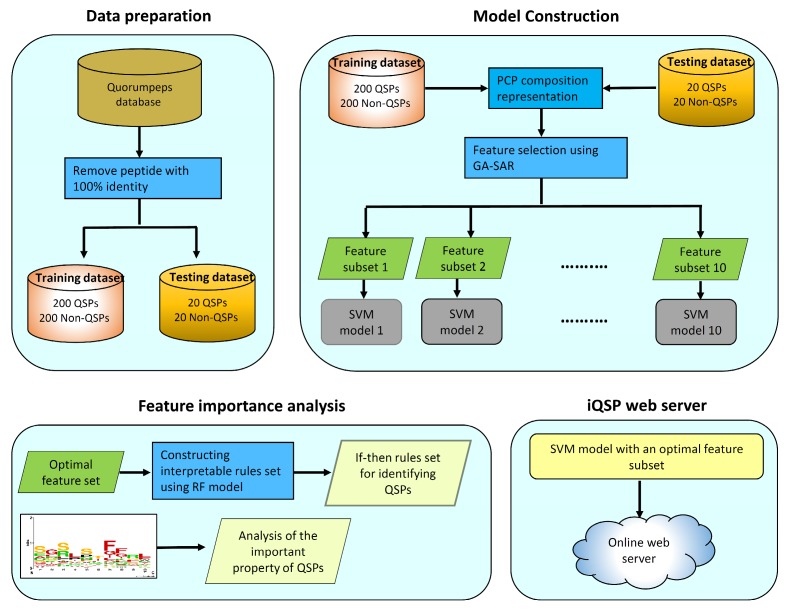
Schematic framework of iQSP. Arrows in the figure represents the direction that data flows from one process to the next process.

**Figure 3 ijms-21-00075-f003:**
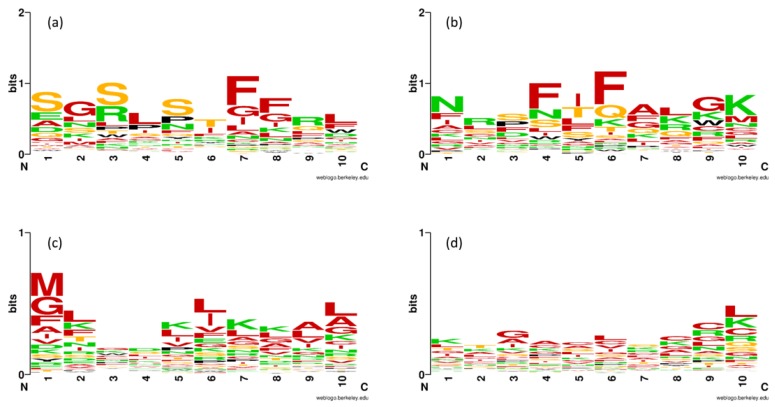
Sequence logo representations, where x- and y-axis represent the first and last 10 residues at N- and C-terminal regions from QSPs (**a**,**b**) and Non-QSPs (**c**,**d**), and proportional to the propensities of amino acids, respectively. Colors are: red for hydrophobic (A, I, L, M, F, V, C, G), green for charged (R, D, E, K), orange for polar (Q, H, S, T), and black for the remaining amino acids (P, Y, W, N).

**Figure 4 ijms-21-00075-f004:**
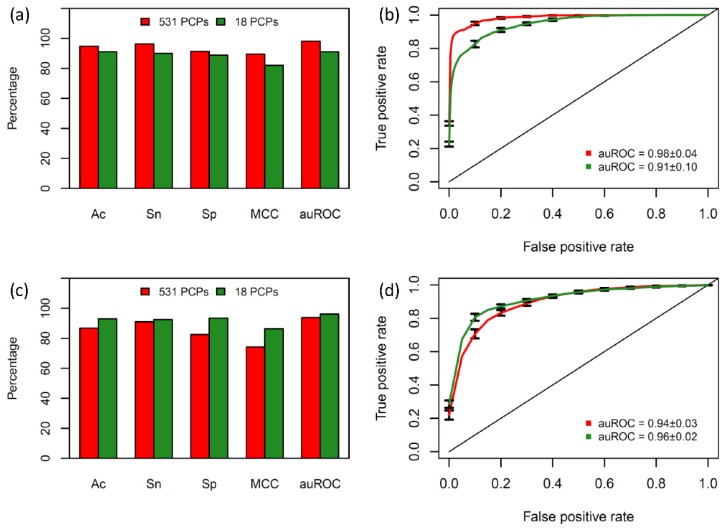
Performance comparisons of SVM models in conjunction with 531 PCPs and the eighteen informative PCPs assessed by 10-fold cross-validation (**a**,**b**) and independent validation test (**c**,**d**).

**Figure 5 ijms-21-00075-f005:**
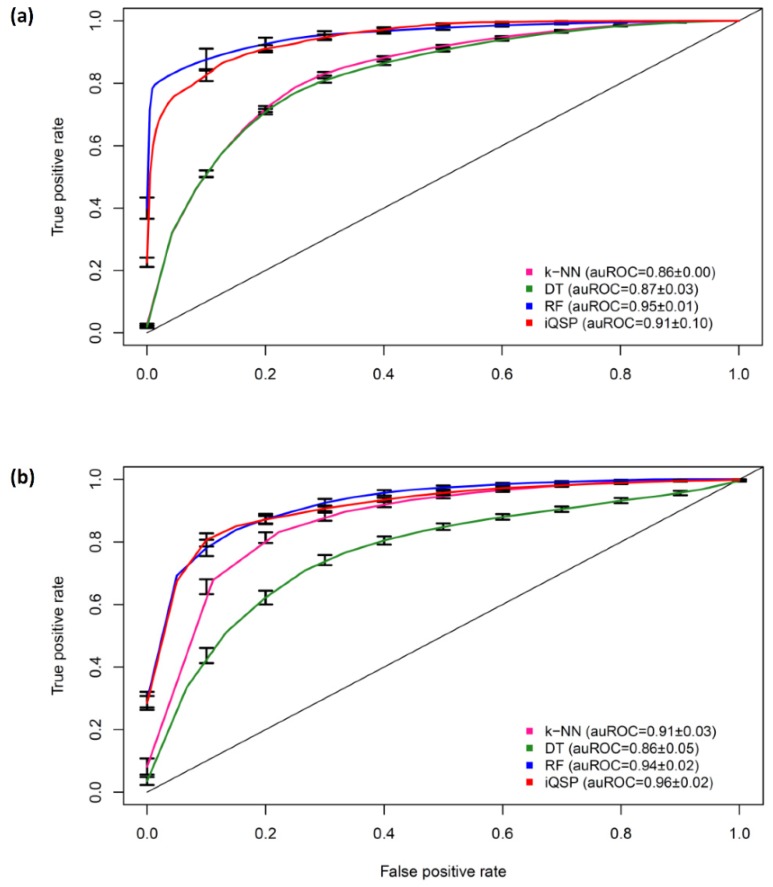
ROC curves of the proposed model iQSP with the conventional classifiers evaluated by 10-fold cross-validation (**a**) and the independent validation test (**b**) with ten independent rounds, where the bar represents the standard deviation of prediction results from ten independent round.

**Figure 6 ijms-21-00075-f006:**
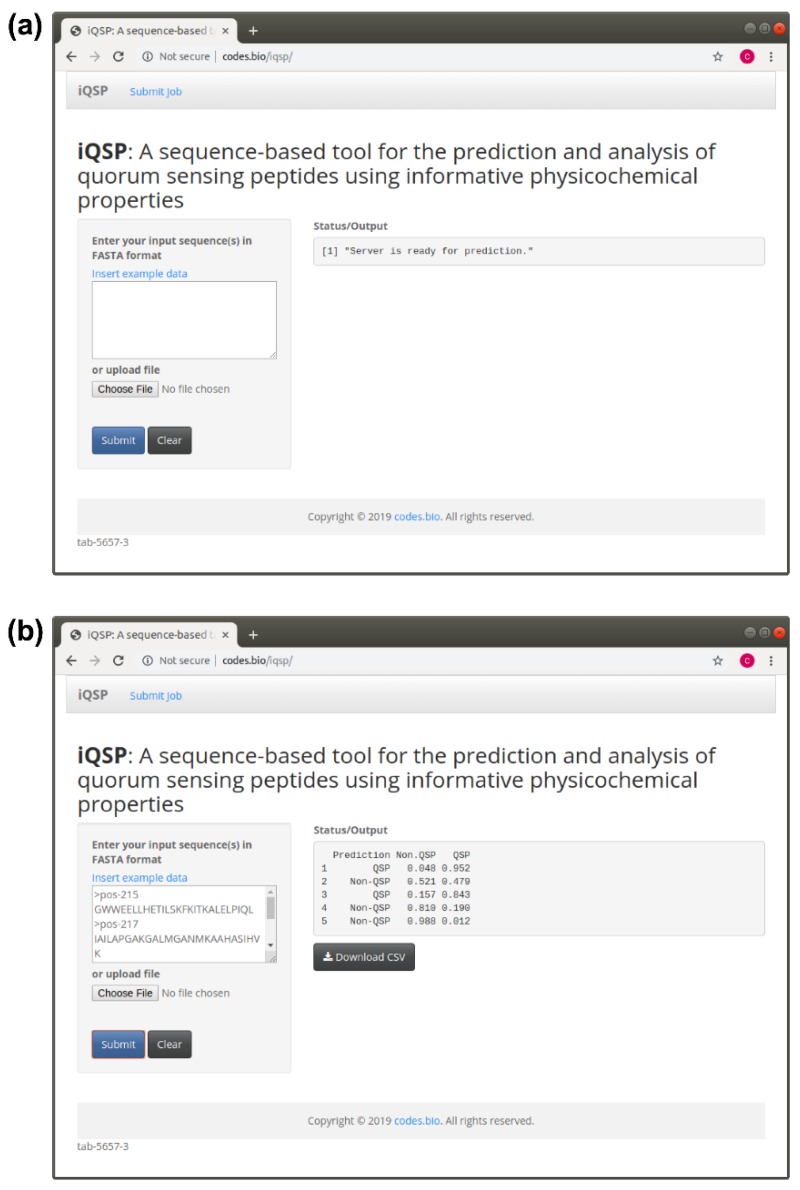
Screenshot of the iQSP web server before (**a**) and after (**b**) submission of the input peptide sequence.

**Table 1 ijms-21-00075-t001:** Summary of existing methods for predicting quorum sensing peptides.

Method	Classifier ^a^	Sequence Feature ^b^	No. of Features	Independent Test
QSPpred	SVM	AAC, DPC, N5C5Bin, PCP	630	Yes
QSPpred-FL	RF	GDP, OVP, CTD, ASDC	913	No
iQSP (this study)	SVM	PCP	18	Yes

^a^ RF: random forest, SVM: support vector machine. ^b^ AAC: amino acid composition, ASDC: adaptive skip dipeptide composition, DPC: dipeptide composition, CTD: composition–transition–distribution, GDP: g-gap dipeptide composition, NCBin: binary pattern of N- and C-terminal residues, OVP: overlapping property features, PCP: physicochemical properties.

**Table 2 ijms-21-00075-t002:** Amino acid compositions (%) of quorum sensing (QSP) and non-quorum sensing (Non-QSP) peptides along with their mean decrease of Gini index (MDGI) values.

Amino Acid	QSP (%)	Non-QSP (%)	Difference	*p*-Value	MDGI
**F**	0.109	0.049	0.059 (1)	0.000	41.94 (1)
**K**	0.047	0.093	−0.045 (20)	0.000	19.37 (2)
**L**	0.076	0.106	−0.030 (18)	0.002	17.57 (3)
**V**	0.043	0.054	−0.011 (14)	0.114	15.56 (4)
**W**	0.034	0.015	0.019 (4)	0.000	10.99 (5)
**I**	0.063	0.067	−0.004 (11)	0.629	10.15 (6)
**A**	0.053	0.084	−0.032 (19)	0.000	9.15 (7)
**Y**	0.042	0.020	0.022 (3)	0.000	8.69 (8)
**R**	0.039	0.050	−0.011 (15)	0.128	7.99 (9)
**C**	0.049	0.061	−0.013 (16)	0.091	7.82 (10)
**S**	0.079	0.062	0.017 (5)	0.023	6.70 (11)
**G**	0.078	0.094	−0.016 (17)	0.032	6.56 (12)
**P**	0.041	0.043	−0.001 (10)	0.846	6.25 (13)
**N**	0.070	0.043	0.026 (2)	0.009	5.24 (14)
**T**	0.051	0.041	0.010 (7)	0.098	4.38 (15)
**D**	0.033	0.029	0.004 (8)	0.411	4.37 (16)
**E**	0.026	0.030	−0.004 (12)	0.423	3.64 (17)
**M**	0.028	0.016	0.012 (6)	0.013	2.84 (18)
**H**	0.010	0.017	−0.007 (13)	0.048	2.71 (19)
**Q**	0.029	0.026	0.003 (9)	0.484	2.49 (20)

**Table 3 ijms-21-00075-t003:** Performance comparisons of SVM models built with various subsets of physicochemical properties evaluated by means of ten-fold cross-validation subjected to ten rounds of random splits.

Subset	# Feature	Ac (%)	Sn (%)	Sp (%)	MCC	AUC
1	14	91.23 ± 1.31	91.17 ± 2.65	88.23 ± 3.03	0.82 ± 0.03	0.95 ± 0.05
2	17	90.69 ± 1.26	92.08 ± 2.82	87.04 ± 2.89	0.82 ± 0.03	0.95 ± 0.05
3	16	91.58 ± 1.77	92.79 ± 2.73	88.45 ± 2.97	0.83 ± 0.04	0.94 ± 0.05
4	17	91.63 ± 1.75	91.49 ± 2.73	89.55 ± 3.71	0.84 ± 0.04	0.92 ± 0.05
5	17	92.01 ± 1.40	91.43 ± 3.64	90.27 ± 2.39	0.84 ± 0.03	0.92 ± 0.08
6	18	91.07 ± 1.77	90.06 ± 2.69	88.79 ± 3.40	0.82 ± 0.04	0.91 ± 0.10
7	15	89.28 ± 1.99	88.78 ± 3.88	86.44 ± 2.67	0.79 ± 0.04	0.93 ± 0.07
8	17	88.24 ± 2.15	85.04 ± 3.61	86.81 ± 3.86	0.76 ± 0.05	0.92 ± 0.07
9	18	90.54 ± 1.26	90.60 ± 4.12	87.57 ± 2.68	0.81 ± 0.03	0.93 ± 0.09
10	17	92.19 ± 1.09	90.12 ± 2.98	92.16 ± 1.96	0.84 ± 0.02	0.93 ± 0.07

# Feature represents the number of features used for constructing a model.

**Table 4 ijms-21-00075-t004:** Performance comparisons of SVM models built with various subsets of physicochemical properties evaluated by means of independent validation test subjected to ten rounds of random splits.

Subset	# Feature	Ac (%)	Sn (%)	Sp (%)	MCC	AUC
1	14	92.50 ± 1.67	95.50 ± 3.69	89.50 ± 4.97	0.85 ± 0.03	0.95 ± 0.03
2	17	91.50 ± 3.58	98.00 ± 3.50	85.00 ± 7.07	0.84 ± 0.07	0.96 ± 0.03
3	16	92.00 ± 2.58	92.50 ± 5.40	91.50 ± 5.80	0.84 ± 0.05	0.95 ± 0.02
4	17	92.25 ± 2.19	93.00 ± 4.83	91.50 ± 4.12	0.85 ± 0.04	0.97 ± 0.01
5	17	92.50 ± 2.36	94.00 ± 5.68	91.00 ± 6.15	0.86 ± 0.05	0.96 ± 0.02
6	18	93.00 ± 1.97	92.50 ± 5.40	93.50 ± 4.12	0.86 ± 0.04	0.96 ± 0.02
7	15	92.00 ± 1.97	94.00 ± 5.16	90.00 ± 7.45	0.85 ± 0.04	0.96 ± 0.02
8	17	91.75 ± 2.90	94.00 ± 5.16	89.50 ± 6.43	0.84 ± 0.06	0.95 ± 0.04
9	18	91.50 ± 3.38	90.50 ± 7.62	92.50 ± 7.91	0.84 ± 0.06	0.97 ± 0.04
10	17	92.50 ± 1.67	95.00 ± 3.33	90.00 ± 4.08	0.85 ± 0.03	0.95 ± 0.04

# Feature represents the number of features used for constructing a model.

**Table 5 ijms-21-00075-t005:** Performance comparisons between iQSP and existing methods assessed by 10-fold cross-validation and independent validation tests.

Method	# Feature	10-Fold CV			Independent Test		
		Ac (%)	MCC	auROC	Ac (%)	MCC	auROC
QSPpred ^a^	430	91.25	0.830	0.960	90.00	0.800	0.950
QSPpred-FL ^b^	913	94.30	0.885	N/A	92.50	0.860	0.962
iQSP	18	91.07 ± 1.77	0.82 ± 0.04	0.91 ± 0.10	93.00 ± 1.97	0.86 ± 0.04	0.96 ± 0.02

^a^ Results were reported from the work of QSPpred. ^b^ Results were obtained by feeding the peptide sequences in the independent set to the webserver of QSPpred-FL. # Feature represents the number of features used for constructing a model. N/A symbol represents the authors did not provide the result.

**Table 6 ijms-21-00075-t006:** Performance comparison of iQSP with other conventional classifiers by using the optimal feature subset. Models were evaluated by means of 10-fold cross-validation and independent validation test subjected to ten rounds of random splits.

Classifier	10-Fold CV			Independent Test		
	Ac (%)	MCC	auROC	Ac (%)	MCC	auROC
*k*-NN	85.13 ± 0.27	0.72 ± 0.01	0.86 ± 0.00	85.75 ± 1.21	0.73 ± 0.02	0.91 ± 0.03
DT	83.57 ± 2.74	0.67 ± 0.06	0.87 ± 0.03	83.75 ± 3.39	0.68 ± 0.07	0.86 ± 0.05
RF	87.93 ± 0.48	0.76 ± 0.01	0.95 ± 0.01	91.00 ± 3.16	0.82 ± 0.06	0.94 ± 0.02
iQSP	91.07 ± 1.77	0.82 ± 0.04	0.91 ± 0.10	93.00 ± 1.97	0.86 ± 0.04	0.96 ± 0.02

**Table 7 ijms-21-00075-t007:** The eighteen informative physicochemical properties [58] derived from the genetic algorithm utilizing self-assessment-report (GA-SAR) algorithm and their MDGI.

Rank	AAindex ID	MDGI	Description
1	QIAN880137	32.55	Weights for coil at the window position of 4 (Qian-Sejnowski, 1988)
2	AURR980102	16.62	Normalized positional residue frequency at helix termini N’ (Aurora-Rose, 1998)
3	ROBB760113	13.56	Information measure for loop (Robson-Suzuki, 1976)
4	PRAM820101	12.62	Intercept in regression analysis (Prabhakaran-Ponnuswamy, 1982)
5	GRAR740101	12.26	Composition (Grantham, 1974)
6	PALJ810111	11.71	Normalized frequency of beta-sheet in alpha + beta class (Palau et al., 1981)
7	PONP800102	10.96	Average gain in surrounding hydrophobicity (Ponnuswamy et al., 1980)
8	MUNV940103	9.07	Free energy in beta-strand conformation (Munoz-Serrano, 1994)
9	DAYM780101	8.64	Amino acid composition (Dayhoff et al., 1978a)
10	MANP780101	8.36	Average surrounding hydrophobicity (Manavalan-Ponnuswamy, 1978)
11	KUMS000103	8.23	Distribution of amino acid residues in the alpha-helices in thermophilic proteins (Kumar et al., 2000)
12	ROBB760104	8.18	Information measure for C-terminal helix (Robson-Suzuki, 1976)
13	ISOY800107	8.09	Normalized relative frequency of double bend (Isogai et al., 1980)
14	GEIM800101	7.80	Alpha-helix indices (Geisow-Roberts, 1980)
15	PRAM900102	7.59	Relative frequency in alpha-helix (Prabhakaran, 1990)
16	NADH010104	7.20	Hydropathy scale based on self-information values in the two-state model (20% accessibility) (Naderi-Manesh et al., 2001)
17	FUKS010106	6.47	Interior composition of amino acids in intracellular proteins of mesophiles (percent) (Fukuchi-Nishikawa, 2001)
18	WIMW960101	5.54	Free energies of transfer of AcWl-X-LL peptides from bilayer interface to water (Wimley-White, 1996)

**Table 8 ijms-21-00075-t008:** Fourteen if–then rules for the prediction of quorum sensing peptides using random forest and the 18 informative physicochemical properties.

No.	Rule	Cover Samples	Misclassified Samples	Ac (%)
1	GRAR740101 ≤ 0.9055 & MANP780101 > 0.7495 & PRAM900102 > 0.848 & QIAN880137 ≤ 0.237	10	0	100.00
2	PONP800102 > −0.751 & PONP800102 ≤ 1.0025 & QIAN880137 ≤ −0.104 & ROBB760104 ≤ 0.3645 & ROBB760104 > −0.5205	61	1	98.36
3	PALJ810111 ≤ 1.369 & QIAN880137 > −0.104 & QIAN880137 ≤ 0.417 & ROBB760113 ≤ 0.5975 & AURR980102 ≤ 0.6955	21	2	90.48
4	GEIM800101 > −0.3135 & GRAR740101 > −0.176 & ISOY800107 ≤ 1.367 & MANP780101 > −0.3325 & PALJ810111 ≤ 1.0905 & QIAN880137 ≤ −0.0985	94	6	93.62
5	PALJ810111 > −0.786 & QIAN880137 > 0.237 & QIAN880137 > 0.403 & ROBB760113 > 0.5975 & AURR980102 ≤ 0.811 & KUMS000103 ≤ 0.793	45	7	84.44
6	GRAR740101 ≤ 0.341 & ISOY800107 ≤ −0.089 & PALJ810111 ≤ 1.2455 & QIAN880137 > −0.009 & ROBB760113 ≤ −0.0285	17	3	82.35
7	GRAR740101 > −0.708 & PRAM900102 ≤ 1.2985 & QIAN880137 > −0.104 & QIAN880137 ≤ 1.105 & AURR980102 ≤ 0.6585 & KUMS000103 ≤ 0.974	94	28	70.21
8	PONP800102 ≤ 1.1095 & PRAM900102 ≤ 0.8295 & QIAN880137 ≤ 0.2625 & QIAN880137 > −0.9055 & ROBB760113 > −0.5875 & ROBB760113 ≤ 1.031	121	15	87.60

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
