# Peer review of "iQSP: A Sequence-Based Tool for the Prediction and Analysis of Quorum Sensing Peptides Using Informative Physicochemical Properties"

_ijms, 2019, doi:10.3390/ijms21010075_

Round 1

Reviewer 1 Report

iQSP: A Sequence-Based Tool for the Prediction and Analysis of Quorum Sensing Peptides Using Informative Physicochemical Properties

Watshara Shoombuatong 1,*, Phasit Charoenkwan 2, Nalini Schaduangrat 1, Theeraphon Piacham and Chanin Nantasenamat

This is an interesting paper because it is directly relevant to a fundamental problem. In view of this, it certainly deserves publication. But to meet the increasingly high-quality standard of the Journal, a compulsory major revision is absolutely needed according to the following points.

(1) A series of recent studies have demonstrated that a lot of useful information for drug development can be obtained by conducting various studies, either experimentally or theoretically. However, different targets would need different approaches. To find effective inhibitors against HIV/AIDS, the Chou’s distorted key theory was applied as briefed in a Wikipedia article at http://en.wikipedia.org/wiki/Chou’s_distorted_key_theory_for_peptide_drugs. For studying drug-binding mechanism or conducting mutagenesis [1, 2], the approach of structural bioinformatics is needed. For studying prion diseases [3] and helix-helix interactions in proteins [4, 5], the wenxiang diagrams or graphs [6, 7] were used. For studying the kinetics of drug metabolism systems, the Chou’s graphic rule was used [8]. For the classification of proteins and its applications in drug development [9], identifying nuclear receptors [10] and [11] as well as analyzing various cellular networking interactions of drugs with different kinds of proteins [12], such as enzymes [13], GPCRs (G protein-coupled receptors) [14], and ion channels [15], the approach of pseudo amino acid composition [16, 17] or Chou’s PseAAC [11] was adopted. For detecting remote homology proteins, the protein profile representation approach was used [18, 19]. For identifying recombination spots of DNAs [20] and their nucleosomal positions [21], the concept of pseudo dinucleotide composition was developed. For identifying the subcellular locations of multiplex proteins [22-29], for finding antimicrobial peptides and their functional types [30], and for classifying drugs according to the ATC (Anatomical Therapeutic Chemical) system recommended by the World Health Organization [31, 32], the multi-label approach [33] was used since each of constituent molecules in these systems possesses one or more than one function or feature.

(2) To make the structure of this paper clearer and easier for readers to follow, the authors should in the end of the Introduction (or right before the beginning of describing their own method) add the following: “As demonstrated by a series of recent publications [34-48] and summarized in two comprehensive review papers  [32, 49], to develop a really useful predictor for a biological system, one needs to follow Chou’s 5-steps rule to go through the following five steps: (1) select or construct a valid benchmark dataset to train and test the predictor; (2) represent the samples with an effective formulation that can truly reflect their intrinsic correlation with the target to be predicted; (3) introduce or develop a powerful algorithm to conduct the prediction; (4) properly perform cross-validation tests to objectively evaluate the anticipated prediction accuracy; (5) establish a user-friendly web-server for the predictor that is accessible to the public. Papers presented for developing a new sequence-analyzing method or statistical predictor by observing the guidelines of Chou’s 5-step rules have the following notable merits: (1) crystal clear in logic development, (2) completely transparent in operation, (3) easily to repeat the reported results by other investigators, (4) with high potential in stimulating other sequence-analyzing methods, and (5) very convenient to be used by the majority of experimental scientists.”  Below, let us elaborate how to deal with these five steps. To learn the importance of the 5-steps rule and how to use it in your own paper, see an insightful Wikipedia article by clicking the link https://en.wikipedia.org/wiki/5-step_rules.

(3) To make the title of this paper more consistent and harmonic with the above suggestion, it should be accordingly changed to: “Use Chou’s 5-steps rule to study …”, which is much more accurate, attractive, and stimulating as well. Actually the “method” used by the authors has been covered by the step3 of the Chou’s 5-steps rule [49].

(4) One of the cornerstones in this study is about feature extraction. But all the features extracted in this paper can be covered by a very powerful web-server called “Pse-in-One” [50] and its updated version “Pse-in-One2.0”, as clearly elucidated very recently [51].  Therefore, to provide the readership with an updated background about using feature extraction to conduct sequence analysis, the authors should in the relevant context  add a prelude such as: “With the explosive growth of biological sequences in the post-genomic era, one of the most important but also most difficult problems in computational biology is how to express a biological sequence with a discrete model or a vector, yet still keep considerable sequence-order information or key pattern characteristic. This is because all the existing machine-learning algorithms can only handle vector but not sequence samples, as elucidated in a comprehensive review [52]. However, a vector defined in a discrete model may completely lose all the sequence-pattern information.  To avoid completely losing the sequence-pattern information for proteins, the pseudo amino acid composition [16] or PseAAC [17] was proposed. Ever since the concept of Chou’s PseAAC was proposed, it has been widely used in nearly all the areas of computational proteomics (see, e.g., [53] [54, 55] as well as a long list of references cited in [56]).  Because it has been widely and increasingly used, recently three powerful open access soft-wares, called ‘PseAAC-Builder’ [57], ‘propy’ [58], and ‘PseAAC-General’ [59], were established: the former two are for generating various modes of Chou’s special PseAAC [60]; while the 3rd one for those of Chou’s general PseAAC [49], including not only all the special modes of feature vectors for proteins but also the higher level feature vectors such as “Functional Domain” mode (see Eqs.9-10 of [49]), “Gene Ontology” mode (see Eqs.11-12 of [49]), and “Sequential Evolution” or “PSSM” mode (see Eqs.13-14 of [49]). Encouraged by the successes of using PseAAC to deal with protein/peptide sequences, the concept of PseKNC (Pseudo K-tuple Nucleotide Composition) [61] was developed for generating various feature vectors for DNA/RNA sequences [62-64] that have proved very useful as well . Particularly, recently a very powerful web-server called ‘Pse-in-One’ [50] and its updated version ‘Pse-in-One2.0’ [51] have been established that can be used to generate any desired feature vectors for protein/peptide and DNA/RNA sequences according to the need of users’ studies.  Also, the current title is not informative and lacking of novelty since the method used by the authors is nothing new but a routine tool.  This has further indicated the necessity and a “Must” to change the title of this paper as suggested in the above Comment 3.

(5) The authors should learn how to write an influential paper and increase its attraction by adding the following statements or discussion in the “Conclusions and Perspective” section:  “It has not escaped our notice that stimulated by the eight master pieces of pioneering papers from the then Chairman of Nobel Prize Committee Sture Forsen [65-72], many follow-up papers have been published [4, 7, 8, 73-112]. They are very useful for in-depth investigation into the topic of the current paper, and we will use them in our future efforts.

(6) It would be highly appreciated if the authors could provide a web-server to display their findings in a flexible way; i.e., by the web-server, users can manipulate the display as desired. It would certainly be very useful for drug design. If the authors couldn’t do that now, to attract the readership to the authors’ future work and to the journal as well, the authors should add a statement in the end of the MS, such as: “As pointed out in [56, 113] and demonstrated in a series of recent publications  (see, e.g., [32, 48, 114-138]) in demonstrating new findings or approaches, user-friendly and publicly accessible web-servers will significantly enhance their impacts [32, 52], driving medicinal chemistry into an unprecedented revolution [32, 56],  we shall make efforts in our future work to provide a web-server to display the findings that can be manipulated by users according to their need.”

(7) To underscore the remarkable and awesome role of the “5-steps rule” in driving proteome/genome analyses and drug development, see a series of recent papers [32, 48, 139-147] where the rule and its wide applications have been very impressively presented from various aspects or at different angles.

To Editor: One of the big problems is that the authors missed too many publications by the previous investigators on the same or closely relevant topics as detailed in the “Comments to the Author”.

REFERENCES

[1] K.C. Chou, D. Jones, R.L. Heinrikson, Prediction of the tertiary structure and substrate binding site of caspase-8. FEBS Letters 419 (1997) 49-54.

[2] K.C. Chou, A.G. Tomasselli, R.L. Heinrikson, Prediction of the Tertiary Structure of a Caspase-9/Inhibitor Complex. FEBS Letters 470 (2000) 249-256.

[3] G.P. Zhou, R.B. Huang, The pH-Triggered Conversion of the PrP(c) to PrP(sc.). Curr Top Med Chem 13 (2013) 1152-63.

[4] G.P. Zhou, The disposition of the LZCC protein residues in wenxiang diagram provides new insights into the protein-protein interaction mechanism. Journal of Theoretical Biology 284 (2011) 142-148.

[5] G.P. Zhou, The Structural Determinations of the Leucine Zipper Coiled-Coil Domains of the cGMP-Dependent Protein Kinase I alpha and its Interaction with the Myosin Binding Subunit of the Myosin Light Chains Phosphase. Proteins & Peptide Letters 18 (2011) 966-978.

[6] K.C. Chou, C.T. Zhang, G.M. Maggiora, Disposition of amphiphilic helices in heteropolar environments. PROTEINS: Structure, Function, and Genetics 28 (1997) 99-108.

[7] K.C. Chou, W.Z. Lin, X. Xiao, Wenxiang: a web-server for drawing wenxiang diagrams Natural Science 3 (2011) 862-865

[8] K.C. Chou, Graphic rule for drug metabolism systems. Current Drug Metabolism 11 (2010) 369-378.

[9] X. Xiao, W.Z. Lin, K.C. Chou, Recent advances in predicting protein classification and their applications to drug development. Current Topics in Medicinal Chemistry 13 (2013) 1622-35.

[10] X. Xiao, P. Wang, K.C. Chou, Recent progresses in identifying nuclear receptors and their families. Current Topics in Medicinal Chemistry 13 (2013) 1192-200.

[11] S.X. Lin, J. Lapointe, Theoretical and experimental biology in one —A symposium in honour of Professor Kuo-Chen Chou’s 50th anniversary and Professor Richard Giegé’s 40th anniversary of their scientific careers. J. Biomedical Science and Engineering (JBiSE) 6 (2013) 435-442.

[12] X. Xiao, J.L. Min, P. Wang, K.C. Chou, Predict drug-protein interaction in cellular networking. Current Topics in Medicinal Chemistry 13 (2013) 1707-12.

[13] J.L. Min, X. Xiao, K.C. Chou, iEzy-Drug: A web server for identifying the interaction between enzymes and drugs in cellular networking. BioMed Research International  (BMRI) 2013 (2013) 701317.

[14] X. Xiao, J.L. Min, P. Wang, K.C. Chou, iGPCR-Drug: A web server for predicting interaction between GPCRs and drugs in cellular networking. PLoS ONE 8 (2013) e72234.

[15] X. Xiao, J.L. Min, P. Wang, K.C. Chou, iCDI-PseFpt: Identify the channel-drug interaction in cellular networking with PseAAC and molecular fingerprints. Journal of Theoretical Biology 337C (2013) 71-79.

[16] K.C. Chou, Prediction of protein cellular attributes using pseudo amino acid composition. PROTEINS: Structure, Function, and Genetics (Erratum: ibid., 2001, Vol.44, 60) 43 (2001) 246-255.

[17] K.C. Chou, Using amphiphilic pseudo amino acid composition to predict enzyme subfamily classes. Bioinformatics 21 (2005) 10-19.

[18] B. Liu, X. Wang, Q. Zou, Q. Dong, Q. Chen, Protein remote homology detection by combining Chou's pseudo amino acid composition and profile-based protein representation. Molecular Informatics 32 (2013) 775-782.

[19] B. Liu, D. Zhang, R. Xu, J. Xu, X. Wang, Q. Chen, Q. Dong, K.C. Chou, Combining evolutionary information extracted from frequency profiles with sequence-based kernels for protein remote homology detection. Bioinformatics 30 (2014) 472-479.

[20] W. Chen, P.M. Feng, H. Lin, K.C. Chou, iRSpot-PseDNC: identify recombination spots with pseudo dinucleotide composition Nucleic Acids Research   41 (2013) e68.

[21] W. Chen, H. Lin, P.M. Feng, C. Ding, Y.C. Zuo, K.C. Chou, iNuc-PhysChem: A Sequence-Based Predictor for Identifying Nucleosomes via Physicochemical Properties. PLoS ONE 7 (2012) e47843.

[22] K.C. Chou, Z.C. Wu, X. Xiao, iLoc-Euk: A Multi-Label Classifier for Predicting the Subcellular Localization of Singleplex and Multiplex Eukaryotic Proteins. PLoS One 6 (2011) e18258.

[23] W.Z. Lin, J.A. Fang, X. Xiao, K.C. Chou, iLoc-Animal: A multi-label learning classifier for predicting subcellular localization of animal proteins Molecular BioSystems 9 (2013) 634-644.

[24] P. Wang, X. Xiao, K.C. Chou, NR-2L: A Two-Level Predictor for Identifying Nuclear Receptor Subfamilies Based on Sequence-Derived Features. PLoS ONE 6 (2011) e23505.

[25] Z.C. Wu, X. Xiao, K.C. Chou, iLoc-Plant: a multi-label classifier for predicting the subcellular localization of plant proteins with both single and multiple sites. Molecular BioSystems 7 (2011) 3287-3297.

[26] Z.C. Wu, X. Xiao, K.C. Chou, iLoc-Gpos: A Multi-Layer Classifier for Predicting the Subcellular Localization of Singleplex and Multiplex Gram-Positive Bacterial Proteins. Protein & Peptide Letters 19 (2012) 4-14.

[27] X. Xiao, Z.C. Wu, K.C. Chou, A multi-label classifier for predicting the subcellular localization of gram-negative bacterial proteins with both single and multiple sites. PLoS ONE 6 (2011) e20592.

[28] X. Xiao, Z.C. Wu, K.C. Chou, iLoc-Virus: A multi-label learning classifier for identifying the subcellular localization of virus proteins with both single and multiple sites. Journal of Theoretical Biology 284 (2011) 42-51.

[29] K.C. Chou, Z.C. Wu, X. Xiao, iLoc-Hum: Using accumulation-label scale to predict subcellular locations of human proteins with both single and multiple sites. Molecular Biosystems 8 (2012) 629-641.

[30] X. Xiao, P. Wang, W.Z. Lin, J.H. Jia, K.C. Chou, iAMP-2L: A two-level multi-label classifier for identifying antimicrobial peptides and their functional types. Analytical Biochemistry 436 (2013) 168-177.

[31] L. Chen, W.M. Zeng, Y.D. Cai, K.Y. Feng, K.C. Chou, Predicting Anatomical Therapeutic Chemical (ATC) classification of drugs by integrating chemical-chemical interactions and similarities. PLoS ONE 7 (2012) e35254.

[32] K.C. Chou, Advance in predicting subcellular localization of multi-label proteins and its implication for developing multi-target drugs. . Current Medicinal Chemistry 26 (2019) 4918-4943.

[33] K.C. Chou, Some remarks on predicting multi-label attributes in molecular biosystems. Molecular Biosystems 9 (2013) 1092-1100.

[34] M. Awais, W. Hussain, Y.D. Khan, N. Rasool, S.A. Khan, K.C. Chou, iPhosH-PseAAC: Identify phosphohistidine sites in proteins by blending statistical moments and position relative features according to the Chou's 5-step rule and general pseudo amino acid composition. IEEE/ACM Trans Comput Biol Bioinform doi:10.1109/TCBB.2019.2919025 (2019).

[35] X. Du, Y. Diao, H. Liu, S. Li, MsDBP: Exploring DNA-binding Proteins by Integrating Multi-scale Sequence Information via Chou's 5-steps Rule. Journal of Proteome Research 18 (2019) 3119-3132.

[36] A. Ehsan, M.K. Mahmood, Y.D. Khan, O.M. Barukab, S.A. Khan, K.C. Chou, iHyd-PseAAC (EPSV): Identify hydroxylation sites in proteins by extracting enhanced position and sequence variant feature via Chou's 5-step rule and general pseudo amino acid composition. Current Genomics 20 (2019) 124-133.

[37] W. Hussain, S.D. Khan, N. Rasool, S.A. Khan, K.C. Chou, SPalmitoylC-PseAAC: A sequence-based model developed via Chou's 5-steps rule and general PseAAC for identifying S-palmitoylation sites in proteins. Anal Biochem 568 (2019) 14-23.

[38] W. Hussain, Y.D. Khan, N. Rasool, S.A. Khan, K.C. Chou, SPrenylC-PseAAC: A sequence-based model developed via Chou's 5-steps rule and general PseAAC for identifying S-prenylation sites in proteins. J Theor Biol 468 (2019) 1-11.

[39] Z. Ju, S.Y. Wang, Prediction of lysine formylation sites using the composition of k-spaced amino acid pairs via Chou's 5-steps rule and general pseudo components. Genomics doi:10.1016/j.ygeno.2019.05.027 (2019).

[40] M. Kabir, S. Ahmad, M. Iqbal, M. Hayat, iNR-2L: A two-level sequence-based predictor developed via Chou's 5-steps rule and general PseAAC for identifying nuclear receptors and their families. Genomics doi:10.1016/j.ygeno.2019.02.006 (2019).

[41] N.Q.K. Le, iN6-methylat (5-step): identifying DNA N(6)-methyladenine sites in rice genome using continuous bag of nucleobases via Chou's 5-step rule. Mol Genet Genomics doi:10.1007/s00438-019-01570-y (2019).

[42] N.Q.K. Le, E.K.Y. Yapp, Q.T. Ho, N. Nagasundaram, Y.Y. Ou, H.Y. Yeh, iEnhancer-5Step: Identifying enhancers using hidden information of DNA sequences via Chou's 5-step rule and word embedding. Anal Biochem 571 (2019) 53-61.

[43] N.Q.K. Le, E.K.Y. Yapp, Y.Y. Ou, H.Y. Yeh, iMotor-CNN: Identifying molecular functions of cytoskeleton motor proteins using 2D convolutional neural network via Chou's 5-step rule. Anal Biochem 575 (2019) 17-26.

[44] Y. Liang, S. Zhang, Identifying DNase I hypersensitive sites using multi-features fusion and F-score features selection via Chou's 5-steps rule. Biophysical Chemistry 253 (2019) 106227.

[45] Q. Ning, Z. Ma, X. Zhao, dForml(KNN)-PseAAC: Detecting formylation sites from protein sequences using K-nearest neighbor algorithm via Chou's 5-step rule and pseudo components. J Theor Biol 470 (2019) 43-49.

[46] Salman, M. Khan, N. Iqbal, T. Hussain, S. Afzal, K.C. Chou, A two-level computation model based on deep learning algorithm for identification of piRNA and their functions via Chou's 5-steps rule. International Journal of Peptide Research and Therapeutics  (IJPRT) Doi: 10.1007/s10989-019-09887-3 (2019).

[47] M. Tahir, H. Tayara, K.T. Chong, iDNA6mA (5-step rule): Identification of DNA N6-methyladenine sites in the rice genome by intelligent computational model via Chou's 5-step rule. CHEMOLAB 189 (2019) 96-101.

[48] K.C. Chou, Progresses in predicting post-translational modification. International Journal of Peptide Research and Therapeutics  (IJPRT) DOI: 10.1007/s10989-019-09893-5 (2019).

[49] K.C. Chou, Some remarks on protein attribute prediction and pseudo amino acid composition (50th Anniversary Year Review, 5-steps rule). Journal of Theoretical Biology 273 (2011) 236-247.

[50] B. Liu, F. Liu, X. Wang, J. Chen, L. Fang, K.C. Chou, Pse-in-One: a web server for generating various modes of pseudo components of DNA, RNA, and protein sequences. Nucleic Acids Research 43 (2015) W65-W71.

[51] B. Liu, H. Wu, K.C. Chou, Pse-in-One 2.0: An improved package of web servers for generating various modes of pseudo components of DNA, RNA, and protein sequences. Natural Science 9 (2017) 67-91.

[52] K.C. Chou, Impacts of bioinformatics to medicinal chemistry. Medicinal Chemistry 11 (2015) 218-234.

[53] A. Dehzangi, R. Heffernan, A. Sharma, J. Lyons, K. Paliwal, A. Sattar, Gram-positive and Gram-negative protein subcellular localization by incorporating evolutionary-based descriptors into Chou's general PseAAC. J Theor Biol 364 (2015) 284-294.

[54] M. Behbahani, H. Mohabatkar, M. Nosrati, Analysis and comparison of lignin peroxidases between fungi and bacteria using three different modes of Chou's general pseudo amino acid composition. J Theor Biol 411 (2016) 1-5.

[55] P.K. Meher, T.K. Sahu, V. Saini, A.R. Rao, Predicting antimicrobial peptides with improved accuracy by incorporating the compositional, physico-chemical and structural features into Chou's general PseAAC. Sci Rep 7 (2017) 42362.

[56] K.C. Chou, An unprecedented revolution in medicinal chemistry driven by the progress of biological science. Current Topics in Medicinal Chemistry 17 (2017) 2337-2358.

[57] P. Du, X. Wang, C. Xu, Y. Gao, PseAAC-Builder: A cross-platform stand-alone program for generating various special Chou's pseudo amino acid compositions. Analytical Biochemistry 425 (2012) 117-119.

[58] D.S. Cao, Q.S. Xu, Y.Z. Liang, propy: a tool to generate various modes of Chou's PseAAC. Bioinformatics 29 (2013) 960-962.

[59] P. Du, S. Gu, Y. Jiao, PseAAC-General: Fast building various modes of general form of Chou's pseudo amino acid composition for large-scale protein datasets. International Journal of Molecular Sciences 15 (2014) 3495-3506.

[60] K.C. Chou, Pseudo amino acid composition and its applications in bioinformatics, proteomics and system biology. Current Proteomics 6 (2009) 262-274.

[61] W. Chen, T.Y. Lei, D.C. Jin, H. Lin, K.C. Chou, PseKNC: a flexible web-server for generating pseudo K-tuple nucleotide composition. Analytical Biochemistry 456 (2014) 53-60.

[62] W. Chen, H. Lin, K.C. Chou, Pseudo nucleotide composition or PseKNC: an effective formulation for analyzing genomic sequences. Mol BioSyst 11 (2015) 2620-2634.

[63] B. Liu, F. Yang, D.S. Huang, K.C. Chou, iPromoter-2L: a two-layer predictor for identifying promoters and their types by multi-window-based PseKNC. Bioinformatics 34 (2018) 33-40.

[64] M. Tahir, H. Tayara, K.T. Chong, iRNA-PseKNC(2methyl): Identify RNA 2'-O-methylation sites by convolution neural network and Chou's pseudo components. J Theor Biol 465 (2019) 1-6.

[65] K.C. Chou, S. Forsen, Diffusion-controlled effects in reversible enzymatic fast reaction system: Critical spherical shell and proximity rate constants. Biophysical Chemistry 12 (1980) 255-263.

[66] K.C. Chou, S. Forsen, Graphical rules for enzyme-catalyzed rate laws. Biochemical Journal 187 (1980) 829-835.

[67] K.C. Chou, S. Forsen, G.Q. Zhou, Three schematic rules for deriving apparent rate constants. Chemica Scripta 16 (1980) 109-113.

[68] K.C. Chou, T.T. Li, S. Forsen, The critical spherical shell in enzymatic fast reaction systems. Biophysical Chemistry 12 (1980) 265-269.

[69] T.T. Li, K.C. Chou, S. Forsen, The flow of substrate molecules in fast enzyme-catalyzed reaction systems. Chemica Scripta 16 (1980) 192-196.

[70] K.C. Chou, R.E. Carter, S. Forsen, A new graphical method for deriving rate equations for complicated mechanisms. Chemica Scripta 18 (1981) 82-86.

[71] K.C. Chou, N.Y. Chen, S. Forsen, The biological functions of low-frequency phonons: 2. Cooperative effects. Chemica Scripta 18 (1981) 126-132.

[72] K.C. Chou, S. Forsen, Graphical rules of steady-state reaction systems. Canadian Journal of Chemistry 59 (1981) 737-755.

[73] K.C. Chou, Low-frequency vibrations of helical structures in protein molecules. Biochemical Journal 209 (1983) 573-580.

[74] K.C. Chou, Identification of low-frequency modes in protein molecules. Biochemical Journal 215 (1983) 465-469.

[75] G.P. Zhou, M.H. Deng, An extension of Chou's graphic rules for deriving enzyme kinetic equations to systems involving parallel reaction pathways. Biochemical Journal 222 (1984) 169-176.

[76] K.C. Chou, Biological functions of low-frequency vibrations ( phonons).  3. Helical structures and microenvironment. Biophysical Journal 45 (1984) 881-889.

[77] K.C. Chou, The biological functions of low-frequency phonons. 4. Resonance effects and allosteric transition. Biophysical Chemistry 20 (1984) 61-71.

[78] K.C. Chou, Low-frequency vibrations of DNA molecules. Biochemical Journal 221 (1984) 27-31.

[79] K.C. Chou, Low-frequency motions in protein molecules: beta-sheet and beta-barrel. Biophysical Journal 48 (1985) 289-297.

[80] K.C. Chou, Prediction of a low-frequency mode in bovine pancreatic trypsin inhibitor molecule. International Journal of Biological Macromolecules 7 (1985) 77-80.

[81] K.C. Chou, Y.S. Kiang, The biological functions of low-frequency phonons: 5. A phenomenological theory. Biophysical Chemistry 22 (1985) 219-235.

[82] K.C. Chou, Origin of low-frequency motion in biological macromolecules: A view of recent progress of quasi-continuity model. Biophysical Chemistry 25 (1986) 105-116.

[83] K.C. Chou, The biological functions of low-frequency phonons: 6. A possible dynamic mechanism of allosteric transition in antibody molecules. Biopolymers 26 (1987) 285-295.

[84] K.C. Chou, Review: Low-frequency collective motion in biomacromolecules and its biological functions. Biophysical Chemistry 30 (1988) 3-48.

[85] K.C. Chou, G.M. Maggiora, The biological functions of low-frequency phonons: 7. The impetus for DNA to accommodate intercalators. British Polymer Journal 20 (1988) 143-148.

[86] K.C. Chou, Low-frequency resonance and cooperativity of hemoglobin. Trends in Biochemical Sciences 14 (1989) 212-213.

[87] K.C. Chou, G.M. Maggiora, B. Mao, Quasi-continuum models of twist-like and accordion-like low-frequency motions in DNA. Biophysical Journal 56 (1989) 295-305.

[88] K.C. Chou, Graphic rules in steady and non-steady enzyme kinetics. Journal of Biological Chemistry 264 (1989) 12074-12079.

[89] K.C. Chou, Review: Applications of graph theory to enzyme kinetics and protein folding kinetics. Steady and non-steady state systems. Biophysical Chemistry 35 (1990) 1-24.

[90] I.W. Althaus, J.J. Chou, A.J. Gonzales, M.R. Diebel, K.C. Chou, F.J. Kezdy, D.L. Romero, P.A. Aristoff, W.G. Tarpley, F. Reusser, Steady-state kinetic studies with the non-nucleoside HIV-1 reverse transcriptase inhibitor U-87201E. Journal of Biological Chemistry 268 (1993) 6119-6124.

[91] I.W. Althaus, A.J. Gonzales, J.J. Chou, M.R. Diebel, K.C. Chou, F.J. Kezdy, D.L. Romero, P.A. Aristoff, W.G. Tarpley, F. Reusser, The quinoline U-78036 is a potent inhibitor of HIV-1 reverse transcriptase. Journal of Biological Chemistry 268 (1993) 14875-14880.

[92] I.W. Althaus, J.J. Chou, A.J. Gonzales, M.R. Diebel, K.C. Chou, F.J. Kezdy, D.L. Romero, P.A. Aristoff, W.G. Tarpley, F. Reusser, Kinetic studies with the nonnucleoside HIV-1 reverse transcriptase inhibitor U-88204E. Biochemistry 32 (1993) 6548-6554.

[93] I.W. Althaus, J.J. Chou, A.J. Gonzales, M.R. Diebel, K.C. Chou, F.J. Kezdy, D.L. Romero, P.A. Aristoff, W.G. Tarpley, F. Reusser, Steady-state kinetic studies with the polysulfonate U-9843, an HIV reverse transcriptase inhibitor. Cellular and Molecular Life Science  (Experientia) 50 (1994) 23-28.

[94] I.W. Althaus, J.J. Chou, A.J. Gonzales, M.R. Diebel, K.C. Chou, F.J. Kezdy, D.L. Romero, R.C. Thomas, P.A. Aristoff, W.G. Tarpley, F. Reusser, Kinetic studies with the non-nucleoside human immunodeficiency virus type-1 reverse transcriptase inhibitor U-90152e. Biochemical Pharmacology 47 (1994) 2017-2028.

[95] K.C. Chou, F.J. Kezdy, F. Reusser, Review: Kinetics of processive nucleic acid polymerases and nucleases. Analytical Biochemistry 221 (1994) 217-230.

[96] K.C. Chou, C.T. Zhang, G.M. Maggiora, Solitary wave dynamics as a mechanism for explaining the internal motion during microtubule growth. Biopolymers 34 (1994) 143-153.

[97] I.W. Althaus, K.C. Chou, K.M. Franks, M.R. Diebel, F.J. Kezdy, D.L. Romero, R.C. Thomas, P.A. Aristoff, W.G. Tarpley, F. Reusser, The benzylthio-pyrididine U-31,355, a potent inhibitor of HIV-1 reverse transcriptase. Biochemical Pharmacology 51 (1996) 743-750.

[98] H. Liu, M. Wang, K.C. Chou, Low-frequency Fourier spectrum for predicting membrane protein types. Biochem Biophys Res Commun (BBRC) 336 (2005) 737-739.

[99] G. Gordon, Designed Electromagnetic Pulsed Therapy: Clinical Applications. Journal of Cellular Physiology 212 (2007) 579-582.

[100] J. Andraos, Kinetic plasticity and the determination of product ratios for kinetic schemes leading to multiple products without rate laws: new methods based on directed graphs. Canadian Journal of Chemistry 86 (2008) 342-357.

[101] K.C. Chou, H.B. Shen, FoldRate: A web-server for predicting protein folding rates from primary sequence. The Open Bioinformatics Journal 3 (2009) 31-50

[102] H.B. Shen, J.N. Song, K.C. Chou, Prediction of protein folding rates from primary sequence by fusing multiple sequential features Journal of Biomedical Science and Engineering (JBiSE) 2 (2009) 136-143.

[103] J.F. Wang, K.C. Chou, Insight into the molecular switch mechanism of human Rab5a from molecular dynamics simulations. Biochem Biophys Res Commun (BBRC) 390 (2009) 608-612.

[104] G. Gordon, Extrinsic electromagnetic fields, low frequency (phonon) vibrations, and control of cell function: a non-linear resonance system. Journal of Biomedical Science and Engineering (JBiSE) 1 (2008) 152-156

[105] A. Madkan, M. Blank, E. Elson, K.C. Chou, M.S. Geddis, R. Goodman, Steps to the clinic with ELF EMF Natural Science 1 (2009) 157-165.

[106] P. Lian, D.Q. Wei, J.F. Wang, K.C. Chou, An allosteric mechanism inferred from molecular dynamics simulations on phospholamban pentamer in lipid membranes. PLoS ONE 6 (2011) e18587.

[107] Q.H. Liao, Q.Z. Gao, J. Wei, K.C. Chou, Docking and Molecular Dynamics Study on the Inhibitory Activity of Novel Inhibitors on Epidermal Growth Factor Receptor (EGFR). Medicinal Chemistry 7 (2011) 24-31.

[108] J. Li, D.Q. Wei, J.F. Wang, Z.T. Yu, K.C. Chou, Molecular Dynamics Simulations of CYP2E1. Medicinal Chemistry 8 (2012) 208-221.

[109] J.F. Wang, K.C. Chou, Recent advances in computational studies on influenza a virus m2 proton channel. Mini Reviews in Medicinal Chemistry 12 (2012) 971-978.

[110] T. Zhang, D.Q. Wei, K.C. Chou, A Pharmacophore Model Specific to Active Site of CYP1A2 with a Novel Molecular Modeling Explorer and CoMFA. Medicinal Chemistry 8 (2012) 198-207.

[111] J. Jia, Z. Liu, X. Xiao, K.C. Chou, iPPI-Esml: an ensemble classifier for identifying the interactions of proteins by incorporating their physicochemical properties and wavelet transforms into PseAAC. J Theor Biol 377 (2015) 47-56.

[112] J. Jia, Z. Liu, X. Xiao, B. Liu, K.C. Chou, Identification of protein-protein binding sites by incorporating the physicochemical properties and stationary wavelet transforms into pseudo amino acid composition (iPPBS-PseAAC). J Biomol Struct Dyn (JBSD) 34 (2016) 1946-1961.

[113] K.C. Chou, H.B. Shen, Recent advances in developing web-servers for predicting protein attributes. Natural Science 1 (2009) 63-92

[114] W. Chen, H. Tang, J. Ye, H. Lin, K.C. Chou, iRNA-PseU: Identifying RNA pseudouridine sites Molecular Therapy - Nucleic Acids   5 (2016) e332.

[115] P. Feng, H. Ding, H. Yang, W. Chen, H. Lin, K.C. Chou, iRNA-PseColl: Identifying the occurrence sites of different RNA modifications by incorporating collective effects of nucleotides into PseKNC. Molecular Therapy - Nucleic Acids 7 (2017) 155-163.

[116] B. Liu, L. Fang, F. Liu, X. Wang, J. Chen, K.C. Chou, Identification of real microRNA precursors with a pseudo structure status composition approach. PLoS ONE 10 (2015) e0121501.

[117] B. Liu, L. Fang, R. Long, X. Lan, K.C. Chou, iEnhancer-2L: a two-layer predictor for identifying enhancers and their strength by pseudo k-tuple nucleotide composition. Bioinformatics 32 (2016) 362-369.

[118] W. Chen, P. Feng, H. Yang, H. Ding, H. Lin, K.C. Chou, iRNA-AI: identifying the adenosine to inosine editing sites in RNA sequences. Oncotarget 8 (2017) 4208-4217.

[119] X. Cheng, S.G. Zhao, X. Xiao, K.C. Chou, iATC-mHyb: a hybrid multi-label classifier for predicting the classification of anatomical therapeutic chemicals. Oncotarget 8 (2017) 58494-58503.

[120] W.R. Qiu, S.Y. Jiang, Z.C. Xu, X. Xiao, K.C. Chou, iRNAm5C-PseDNC: identifying RNA 5-methylcytosine sites by incorporating physical-chemical properties into pseudo dinucleotide composition. Oncotarget 8 (2017) 41178-41188.

[121] C.J. Zhang, H. Tang, W.C. Li, H. Lin, W. Chen, K.C. Chou, iOri-Human: identify human origin of replication by incorporating dinucleotide physicochemical properties into pseudo nucleotide composition. Oncotarget 7 (2016) 69783-69793.

[122] J. Jia, Z. Liu, X. Xiao, B. Liu, K.C. Chou, pSuc-Lys: Predict lysine succinylation sites in proteins with PseAAC and ensemble random forest approach. Journal of Theoretical Biology 394 (2016) 223-230.

[123] X. Cheng, S.G. Zhao, X. Xiao, K.C. Chou, iATC-mISF: a multi-label classifier for predicting the classes of anatomical therapeutic chemicals. Bioinformatics  (Corrigendum, ibid., 2017, Vol.33, 2610) 33 (2017) 341-346.

[124] B. Liu, S. Wang, R. Long, K.C. Chou, iRSpot-EL: identify recombination spots with an ensemble learning approach. Bioinformatics 33 (2017) 35-41.

[125] X. Cheng, X. Xiao, K.C. Chou, pLoc-mPlant: predict subcellular localization of multi-location plant proteins via incorporating the optimal GO information into general PseAAC. Molecular BioSystems 13 (2017) 1722-1727.

[126] X. Cheng, X. Xiao, K.C. Chou, pLoc-mVirus: predict subcellular localization of multi-location virus proteins via incorporating the optimal GO information into general PseAAC. Gene (Erratum: ibid., 2018, Vol.644, 156-156) 628 (2017) 315-321.

[127] X. Cheng, S.G. Zhao, W.Z. Lin, X. Xiao, K.C. Chou, pLoc-mAnimal: predict subcellular localization of animal proteins with both single and multiple sites. Bioinformatics 33 (2017) 3524-3531.

[128] X. Xiao, X. Cheng, S. Su, Q. Nao, K.C. Chou, pLoc-mGpos: Incorporate key gene ontology information into general PseAAC for predicting subcellular localization of Gram-positive bacterial proteins. Natural Science 9 (2017) 331-349.

[129] X. Cheng, X. Xiao, K.C. Chou, pLoc-mEuk: Predict subcellular localization of multi-label eukaryotic proteins by extracting the key GO information into general PseAAC. Genomics 110 (2018) 50-58.

[130] X. Cheng, X. Xiao, K.C. Chou, pLoc-mGneg: Predict subcellular localization of Gram-negative bacterial proteins by deep gene ontology learning via general PseAAC. Genomics 110 (2018) 231-239.

[131] X. Cheng, X. Xiao, K.C. Chou, pLoc-mHum: predict subcellular localization of multi-location human proteins via general PseAAC to winnow out the crucial GO information. Bioinformatics 34 (2018) 1448-1456.

[132] X. Cheng, X. Xiao, K.C. Chou, pLoc_bal-mGneg: predict subcellular localization of Gram-negative bacterial proteins by quasi-balancing training dataset and general PseAAC. Journal of Theoretical  Biology 458 (2018) 92-102.

[133] X. Cheng, X. Xiao, K.C. Chou, pLoc_bal-mPlant: predict subcellular localization of plant proteins by general PseAAC and balancing training dataset Curr Pharm Des 24 (2018) 4013-4022.

[134] K.C. Chou, X. Cheng, X. Xiao, pLoc_bal-mHum: predict subcellular localization of human proteins by PseAAC and quasi-balancing training dataset Genomics 111 (2019) 1274-1282.

[135] X. Xiao, X. Cheng, G. Chen, Q. Mao, K.C. Chou, pLoc_bal-mVirus: Predict Subcellular Localization of Multi-Label Virus Proteins by Chou's General PseAAC and IHTS Treatment to Balance Training Dataset. Med Chem 15 (2018) 496-509.

[136] X. Cheng, W.Z. Lin, X. Xiao, K.C. Chou, pLoc_bal-mAnimal: predict subcellular localization of animal proteins by balancing training dataset and PseAAC. Bioinformatics 35 (2019) 398-406.

[137] K.C. Chou, X. Cheng, X. Xiao, pLoc_bal-mEuk: predict subcellular localization of eukaryotic proteins by general PseAAC and quasi-balancing training dataset. Med Chem 15 (2019) 472-485.

[138] X. Xiao, X. Cheng, G. Chen, Q. Mao, K.C. Chou, pLoc_bal-mGpos: predict subcellular localization of Gram-positive bacterial proteins by quasi-balancing training dataset and PseAAC. Genomics 111 (2019) 886-892.

[139] K.C. Chou, Impacts of pseudo amino acid components and 5-steps rule to proteomics and proteome analysis. Current Topics in Medicinak Chemistry (CTMC) (Special Issue ed. G.P Zhou) DOI: 10.2174/1568026619666191018100141 (2019).

[140] K.C. Chou, Two kinds of metrics for computational biology. Genomics doi:10.1016/j.ygeno.2019.08.008 (2019).

[141] K.C. Chou, Proposing pseudo amino acid components is an important milestone for proteome and genome analyses. International Journal for Peptide Research and Therapeutics (IJPRT) DOI: 10.1007/s10989-019-09910-7 (2019).

[142] K.C. Chou, An insightful recollection for predicting protein subcellular locations in multi-label systems. Genomics doi:10.1016/j.ygeno.2019.08.008 (2019).

[143] K.C. Chou, Recent Progresses in Predicting Protein Subcellular Localization with Artificial Intelligence (AI) Tools Developed Via the 5-Steps Rule. Japanese Journal of Gastroenterology and Hepatology Vol.2, doi:www.jjgastrohepto.org (2019).

[144] K.C. Chou, An insightful recollection since the distorted key theory was born about 23 years ago. Genomics doi: 10.1016/j.ygeno.2019.09.001 (2019).

[145] K.C. Chou, Artificial intelligence (AI) tools constructed via the 5-steps rule for predicting post-translational modifications. Trends in Artificial Inttelengence (TIA) 3 (2019) 60-74.

[146] K.C. Chou, Gordon Life Science Institute: Its philosophy, achievements, and perspective. Annals of Cancer Therapy and Pharmacology 2 (2019) 001-26.

[147] K.C. Chou, Distorted Key Theory and Its Implication for Drug Development. Current Genomics 17, doi:10.2174/1570164617666191025101914 (2020).

Author Response

This is an interesting paper because it is directly relevant to a fundamental problem. In view of this, it certainly deserves publication. But to meet the increasingly high-quality standard of the Journal, a compulsory major revision is absolutely needed according to the following points.

Response: Thank you for the encouraging and kind words of inspiration

Point 1 A series of recent studies have demonstrated that a lot of useful information for drug development can be obtained by conducting various studies, either experimentally or theoretically. However, different targets would need different approaches. To find effective inhibitors against HIV/AIDS, the Chou’s distorted key theory was applied as briefed in a Wikipedia article at http://en.wikipedia.org/wiki/Chou’s_distorted_key_theory_for_peptide_drugs. For studying drug-binding mechanism or conducting mutagenesis [1, 2], the approach of structural bioinformatics is needed. For studying prion diseases [3] and helix-helix interactions in proteins [4, 5], the wenxiang diagrams or graphs [6, 7] were used. For studying the kinetics of drug metabolism systems, the Chou’s graphic rule was used [8]. For the classification of proteins and its applications in drug development [9], identifying nuclear receptors [10] and [11] as well as analyzing various cellular networking interactions of drugs with different kinds of proteins [12], such as enzymes [13], GPCRs (G protein-coupled receptors) [14], and ion channels [15], the approach of pseudo amino acid composition [16, 17] or Chou’s PseAAC [11] was adopted. For detecting remote homology proteins, the protein profile representation approach was used [18, 19]. For identifying recombination spots of DNAs [20] and their nucleosomal positions [21], the concept of pseudo dinucleotide composition was developed. For identifying the subcellular locations of multiplex proteins [22-29], for finding antimicrobial peptides and their functional types [30], and for classifying drugs according to the ATC (Anatomical Therapeutic Chemical) system recommended by the World Health Organization [31, 32], the multi-label approach [33] was used since each of constituent molecules in these systems possesses one or more than one function or feature.

Response 1: Thank you for the valuable suggestion. We will keep this point to improve the quality of our future works.

Point 2 To make the structure of this paper clearer and easier for readers to follow, the authors should in the end of the Introduction (or right before the beginning of describing their own method) add the following: “As demonstrated by a series of recent publications [34-48] and summarized in two comprehensive review papers  [32, 49], to develop a really useful predictor for a biological system, one needs to follow Chou’s 5-steps rule to go through the following five steps: (1) select or construct a valid benchmark dataset to train and test the predictor; (2) represent the samples with an effective formulation that can truly reflect their intrinsic correlation with the target to be predicted; (3) introduce or develop a powerful algorithm to conduct the prediction; (4) properly perform cross-validation tests to objectively evaluate the anticipated prediction accuracy; (5) establish a user-friendly web-server for the predictor that is accessible to the public. Papers presented for developing a new sequence-analyzing method or statistical predictor by observing the guidelines of Chou’s 5-step rules have the following notable merits: (1) crystal clear in logic development, (2) completely transparent in operation, (3) easily to repeat the reported results by other investigators, (4) with high potential in stimulating other sequence-analyzing methods, and (5) very convenient to be used by the majority of experimental scientists.”  Below, let us elaborate how to deal with these five steps. To learn the importance of the 5-steps rule and how to use it in your own paper, see an insightful Wikipedia article by clicking the link https://en.wikipedia.org/wiki/5-step_rules.

Response 2: Thank you for the great comment. As suggested, we have added your suggested details in the first paragraph of the Materials and Methods section

Point 3 To make the title of this paper more consistent and harmonic with the above suggestion, it should be accordingly changed to: “Use Chou’s 5-steps rule to study …”, which is much more accurate, attractive, and stimulating as well. Actually the “method” used by the authors has been covered by the step3 of the Chou’s 5-steps rule [49].

Response 3: Thank you for your kind suggestion. In fact, your suggestion is more useful for our current work. However, we have mentioned the Chou’s 5-steps rule (as mentioned in Point 2) in the first paragraph of the Materials and Methods section.

Point 4 One of the cornerstones in this study is about feature extraction. But all the features extracted in this paper can be covered by a very powerful web-server called “Pse-in-One” [50] and its updated version “Pse-in-One2.0”, as clearly elucidated very recently [51].  Therefore, to provide the readership with an updated background about using feature extraction to conduct sequence analysis, the authors should in the relevant context  add a prelude such as: “With the explosive growth of biological sequences in the post-genomic era, one of the most important but also most difficult problems in computational biology is how to express a biological sequence with a discrete model or a vector, yet still keep considerable sequence-order information or key pattern characteristic. This is because all the existing machine-learning algorithms can only handle vector but not sequence samples, as elucidated in a comprehensive review [52]. However, a vector defined in a discrete model may completely lose all the sequence-pattern information.  To avoid completely losing the sequence-pattern information for proteins, the pseudo amino acid composition [16] or PseAAC [17] was proposed. Ever since the concept of Chou’s PseAAC was proposed, it has been widely used in nearly all the areas of computational proteomics (see, e.g., [53] [54, 55] as well as a long list of references cited in [56]).  Because it has been widely and increasingly used, recently three powerful open access soft-wares, called ‘PseAAC-Builder’ [57], ‘propy’ [58], and ‘PseAAC-General’ [59], were established: the former two are for generating various modes of Chou’s special PseAAC [60]; while the 3rd one for those of Chou’s general PseAAC [49], including not only all the special modes of feature vectors for proteins but also the higher level feature vectors such as “Functional Domain” mode (see Eqs.9-10 of [49]), “Gene Ontology” mode (see Eqs.11-12 of [49]), and “Sequential Evolution” or “PSSM” mode (see Eqs.13-14 of [49]). Encouraged by the successes of using PseAAC to deal with protein/peptide sequences, the concept of PseKNC (Pseudo K-tuple Nucleotide Composition) [61] was developed for generating various feature vectors for DNA/RNA sequences [62-64] that have proved very useful as well . Particularly, recently a very powerful web-server called ‘Pse-in-One’ [50] and its updated version ‘Pse-in-One2.0’ [51] have been established that can be used to generate any desired feature vectors for protein/peptide and DNA/RNA sequences according to the need of users’ studies.  Also, the current title is not informative and lacking of novelty since the method used by the authors is nothing new but a routine tool.  This has further indicated the necessity and a “Must” to change the title of this paper as suggested in the above Comment 3.

Response 4: Thank you for the great suggestions. We have added the related detail in the Conclusion section.

Point 5 The authors should learn how to write an influential paper and increase its attraction by adding the following statements or discussion in the “Conclusions and Perspective” section:  “It has not escaped our notice that stimulated by the eight master pieces of pioneering papers from the then Chairman of Nobel Prize Committee Sture Forsen [65-72], many follow-up papers have been published [4, 7, 8, 73-112]. They are very useful for in-depth investigation into the topic of the current paper, and we will use them in our future efforts.

Response 5: Thank you for the valuable comments in helping to improve this manuscript. As suggested, we have added these details in the Conclusion section.

Point 6 It would be highly appreciated if the authors could provide a web-server to display their findings in a flexible way; i.e., by the web-server, users can manipulate the display as desired. It would certainly be very useful for drug design. If the authors couldn’t do that now, to attract the readership to the authors’ future work and to the journal as well, the authors should add a statement in the end of the MS, such as: “As pointed out in [56, 113] and demonstrated in a series of recent publications  (see, e.g., [32, 48, 114-138]) in demonstrating new findings or approaches, user-friendly and publicly accessible web-servers will significantly enhance their impacts [32, 52], driving medicinal chemistry into an unprecedented revolution [32, 56],  we shall make efforts in our future work to provide a web-server to display the findings that can be manipulated by users according to their need.”

Response 6: Thank you for the valuable comments in helping to improve this manuscript. As suggested, we have added it in the 3.7. Construction of the iQSP web server section

Point 7 To underscore the remarkable and awesome role of the “5-steps rule” in driving proteome/genome analyses and drug development, see a series of recent papers [32, 48, 139-147] where the rule and its wide applications have been very impressively presented from various aspects or at different angles.

Response 7: Thank you for the valuable suggestion. We will keep this point to improve the quality of our future works. In addition, we have added the “5-steps rule in the first paragraph of the Materials and Methods section.

Reviewer 2 Report

 Th authors present an interesting methodology for predicting and analyzing QSPs. This is a very hot topic and the presented tool may be very helpful for those working in this field. The manuscript is well written and the conclusions are robust.

Author Response

The authors present an interesting methodology for predicting and analyzing QSPs. This is a very hot topic and the presented tool may be very helpful for those working in this field. The manuscript is well written and the conclusions are robust.

Response: Thank you for the encouraging and kind words of inspiration.

Round 2

Reviewer 1 Report

The authors did a very poor and lousy job in the revised version. The authors should carefully read my comments in the 1st-round Review Report, and do a very careful major revision again. Particularly, the title of this paper must be changed to the one as clearly suggested there, which sounds much more accurate, attractive, and stimulating as well.

The authors’ choice for not using the suggested title is completely wrong.  This is because the “5-steps rule” can be used to analysis of many problems in proteomics/genomics, even including but not limited to, the commercial problems and bank systems, and material science systems, the same must be true for the system of the current study. As for the importance of the 5-steps rule and how to use it, see an insightful Wikipedia article at https://en.wikipedia.org/wiki/5-step_rules. The only difference between the two is how to formulate the statistical samples or events with an effective mathematical expression that can truly reflect their intrinsic correlation with the target to be predicted. It just like the case of many machine-learning algorithms. They can be widely used in nearly all the areas of statistical analysis. If the authors change the title as suggested, this paper can be accepted for publication.

Author Response

The authors did a very poor and lousy job in the revised version. The authors should carefully read my comments in the 1st-round Review Report, and do a very careful major revision again. Particularly, the title of this paper must be changed to the one as clearly suggested there, which sounds much more accurate, attractive, and stimulating as well.

The authors’ choice for not using the suggested title is completely wrong.  This is because the “5-steps rule” can be used to analysis of many problems in proteomics/genomics, even including but not limited to, the commercial problems and bank systems, and material science systems, the same must be true for the system of the current study. As for the importance of the 5-steps rule and how to use it, see an insightful Wikipedia article at https://en.wikipedia.org/wiki/5-step_rules. The only difference between the two is how to formulate the statistical samples or events with an effective mathematical expression that can truly reflect their intrinsic correlation with the target to be predicted. It just like the case of many machine-learning algorithms. They can be widely used in nearly all the areas of statistical analysis. If the authors change the title as suggested, this paper can be accepted for publication.

Response: Thank you for the kind suggestion. As suggested, the title has been changed to be “iQSP: A Sequence-Based Tool for the Prediction and Analysis of Quorum Sensing Peptides Via Chou’s 5-Steps Rule and Informative Physicochemical Properties”.

Round 3

Reviewer 1 Report

The authors did a good job in the 2nd revised version. But the title of this paper should be changed to “Using Chou’s 5-steps rule to predict and analyze quorum sensing peptides", which sounds more pleasant. However, to speed up publication, this change can be done in finalyzing the MS or during the proof stage.